# Differences in regulation mechanisms of glutamine synthetases from methanogenic archaea unveiled by structural investigations

Marie-Caroline Müller[1], Olivier N. Lemaire[1], Julia M. Kurth[2,3], Cornelia U. Welte [2] & Tristan Wagner [1✉]

Glutamine synthetases (GS) catalyze the ATP-dependent ammonium assimilation, the initial step of nitrogen acquisition that must be under tight control to fit cellular needs. While their catalytic mechanisms and regulations are well-characterized in bacteria and eukaryotes, only limited knowledge exists in archaea. Here, we solved two archaeal GS structures and unveiled unexpected differences in their regulatory mechanisms. GS from *Methanothermococcus thermolithotrophicus* is inactive in its resting state and switched on by 2-oxoglutarate, a sensor of cellular nitrogen deficiency. The enzyme activation overlays remarkably well with the reported cellular concentration for 2-oxoglutarate. Its binding to an allosteric pocket reconfigures the active site through long-range conformational changes. The homolog from *Methermicoccus shengliensis* does not harbor the 2-oxoglutarate binding motif and, consequently, is 2-oxoglutarate insensitive. Instead, it is directly feedback-inhibited through glutamine recognition by the catalytic Asp50′-loop, a mechanism common to bacterial homologs, but absent in *M. thermolithotrophicus* due to residue substitution. Analyses of residue conservation in archaeal GS suggest that both regulations are widespread and not mutually exclusive. While the effectors and their binding sites are surprisingly different, the molecular mechanisms underlying their mode of action on GS activity operate on the same molecular determinants in the active site.

[1] Microbial Metabolism Research Group, Max Planck Institute for Marine Microbiology, Celsiusstraße 1, 28359 Bremen, Germany. [2] Department of Microbiology, Radboud Institute for Biological and Environmental Sciences, Radboud University, Heyendaalseweg 135, 6525 AJ Nijmegen, The Netherlands. [3] Present address: Microcosm Earth Center, Philipps-University Marburg and Max Planck Institute for Terrestrial Microbiology, Hans-Meerwein-Str. 4, 35032 Marburg, Germany. ✉email: twagner@mpi-bremen.de

Nitrogen is an essential component of all living cells. Its most reduced state, ammonia (here representative for $NH_3$ and $NH_4^+$ in equilibrium), is one of the most common nitrogen sources assimilated by the microbial world[1–4]. Ammonia enters the central nitrogen metabolism via two systems: the glutamine synthetase – glutamate synthase (GS-GOGAT) couple and/or the glutamate dehydrogenase (GDH). While the GDH carries out the reversible reaction of reductive amination of 2-oxoglutarate (2OG) using NAD(P)H, the couple GS/GOGAT performs an ATP-dependent process[5–8]. Because of its higher affinity for ammonia, the GS-GOGAT couple is notably known to be more effective than the GDH in ammonia-limited environments[1]. The nitrogen assimilation by GS-GOGAT is operated in two steps (Fig. S1a). The GS initially produces glutamine via the condensation of ammonia on glutamate, a reaction coupled to ATP-hydrolysis that requires $Mg^{2+}$ or $Mn^{2+}$[1]. Then, GOGAT performs the deamination of the synthesized glutamine and the transfer of ammonia to 2OG to ultimately form two molecules of glutamate, with the concomitant oxidation of an electron donor (e.g., NADH)[1,7].

GS can be categorized into three types[1,9]. Type I is present in prokaryotes, and some homologs were found in eukaryotes. It organizes as a homo-dodecamer of ~55 kDa-large subunits[10]. Type II is composed of a homo-decameric assembly of ~40 kDa-large subunits and is common in bacteria and eukaryotes[11,12]. Type III is found in bacteria, archaea and eukaryotes and is composed of a homo-dodecamer of ~75 kDa-large subunits[1,13]. The GS type I (GSI) is further subdivided into three classes: (i) the GSI-α found in *Archaea, Actinobacteria, Desulfobacterota* and *Bacillota* (formerly *Firmicutes*), (ii) the GSI-β, present in many bacteria and a few archaea and (iii) the GSI-γ mostly found in bacteria[9]. Studies on the evolution of this ancient enzyme led to different scenarios, explaining the puzzling phylogeny of GS that should be the result of gene deletion and horizontal gene transfer, including that GSI-α and GSI-β was already separated in the Last Universal Common Ancestor and a loss of GSI-α and/or GSI-β occurred in different lineages[9,14–17]. The GSI-α in *Bacillota* was obtained through lateral gene transfer from an archaeal ancestor[9,14]. The structural characterization of some GSI-α and GSI-β unveiled a similar organization[6,18,19]. The dodecamer is organized as two stacked hexameric rings, and each polypeptide is organized into two parts, a shorter N-terminal domain (β-Grasp fold) and the remaining C-terminal domain (Fig. S1b-g). The N-terminal domain is mainly responsible for the ring association, while the C-terminal domain composes the main part of the ring and is responsible for inter-ring interactions. GSI-β differs structurally from GSI-α by a 25 amino acid-long extension involved in inter-ring stabilization (Fig. S1b-d). The active sites are located at the interface of the N- and C-terminal domains of the adjacent subunit in the hexameric ring. Each active site is structured as a "bifunnel" with ATP and glutamate binding on opposite sides. The ATP binding site is usually referred to as the top of the funnel since the opening lies towards the external ring surface, and the glutamate-binding site as the bottom of the funnel. Divalent metal cations ($Mn^{2+}$ or $Mg^{2+}$) are coordinated on the C-terminal domain and positioned at the center of the bifunnel. Two loops in the active center are critical for catalysis: the Glu-flap, involved in shielding the active site during catalysis and deprotonating the intermediate product, and the Asp-50' loop, which binds and deprotonates ammonium. In this concerted orchestra, the glutamate is first phosphorylated on its γ-carboxyl group by the ATP donor, and secondly, the ammonia is incorporated, releasing the products glutamine, ADP and inorganic phosphate[6,20]. GSI-γ are comprised of only the catalytic domain lacking the N-terminus. A majority of biochemically characterized GSI-γ members lack GS biosynthetic activity and instead function as γ-glutamyl-polyamine synthetases[9].

The GSI-α and GSI-β also differ regarding their regulation. The GSI-β have a complex multi-level regulation by adenylylation, a feature not conserved in GSI-α[8]. The regulation of GSI-α in *Bacillus subtilis* was shown to depend on feedback inhibition by the enzyme's product glutamine[18]. The glutamine-inhibited enzyme also binds the regulator GlnR, triggering a switch from a dodecameric GS to an inactive tetradecameric form[19]. The GS-GlnR complex finally acts as a transcription repressor for genes involved in nitrogen assimilation, including the GS.

The archaeal GSI-α has never been structurally characterized, and only sporadic studies exist[21–25]. While the activity of the GSI-α of *Haloferax mediterranei* is inhibited by glutamine[26], other radically different regulations have been discovered. In the same archaeon, the molecule 2OG was shown to be a more efficient regulator with a 12-fold activity stimulation, reaching 18-fold when $P_{II}$-family regulatory proteins $GlnK_1$ or $GlnK_2$ were added in addition to 2OG[26]. 2OG and GlnK also enhance the activity of the enzyme from the methanogen *Methanosarcina mazei*, as well as the 23 residue-large sP26 protein[27,28], although an inhibitory effect of GlnK (under different buffer conditions) has also been shown[28]. The regulatory proteins GlnK and sP26, whose expression is triggered under nitrogen starvation, form a complex and make a tight and specific interaction with the GS[27,29]. The direct control by 2OG is particularly elegant as the metabolite is a cellular sensor for nitrogen deprivation and has been shown to act on several key nodes of nitrogen acquisition in methanogens (Fig. S1a)[8,26–28,30].

To shed light on the regulatory mechanisms of archaeal GSI-α, we performed biochemical and structural characterizations on the enzymes from two thermophilic methanogens belonging to different phyla. Our results illustrate the different regulation systems at the molecular level, including an unforeseen switch-on controlled by 2OG.

## Results

### *Mt*GS activity is strictly dependent on 2OG in contrast to *Ms*GS.

*Methanothermococcus thermolithotrophicus* belonging to the *Methanococcales* order is a strictly hydrogenotrophic methanogen which possesses a single gene coding for a GSI-α (the gene product is referred to as *Mt*GS) while lacking any genes coding for GDH. The purification of *Mt*GS was previously published[31]. We improved the purification protocol to obtain an anaerobically purified *Mt*GS allowing its detailed enzymatic characterization. Denaturing and high-resolution clear native polyacrylamide gel electrophoresis (hrCN PAGE) are coherent with a dodecameric organization of the ~50 kDa peptide (complex size estimated at 540.3 kDa, Fig. 1a and b). The activity of *Mt*GS, measured via a coupled enzyme assay (Fig. S2), could not be detected despite the addition of $Mg^{2+}$ or $Mn^{2+}$, an increase of the protein concentration (up to 0.75 mg.ml$^{-1}$ final concentration) or an extension of the incubation time. As 2OG acts as an activity stimulator in archaeal GS[26–28], 2OG was added to the assay and revealed to be essential for *Mt*GS activity under these experimental conditions (Fig. 1c). No activity was detected below 0.06 mM 2OG, 50% activity was reached at 0.17 mM and saturation occurred above ~0.6 mM. Noteworthy, exposure to oxygen severely reduced the enzyme activity (1 h of exposure to $O_2$ decreased the activity to 6%). Specificity for 2OG was tested by substitution with malate or succinate, which are structurally similar to 2OG. At 15 mM, both surrogates could not activate *Mt*GS. Malate or succinate addition to the 2OG-activated enzyme slightly impaired the activity, an effect that might be due to a competition for the binding site (Fig. S3).

To investigate whether the 2OG dependency is a feature unique to *M. thermolithotrophicus* or also distributed among other methanogens, we purified the GS from the phylogenetically distant

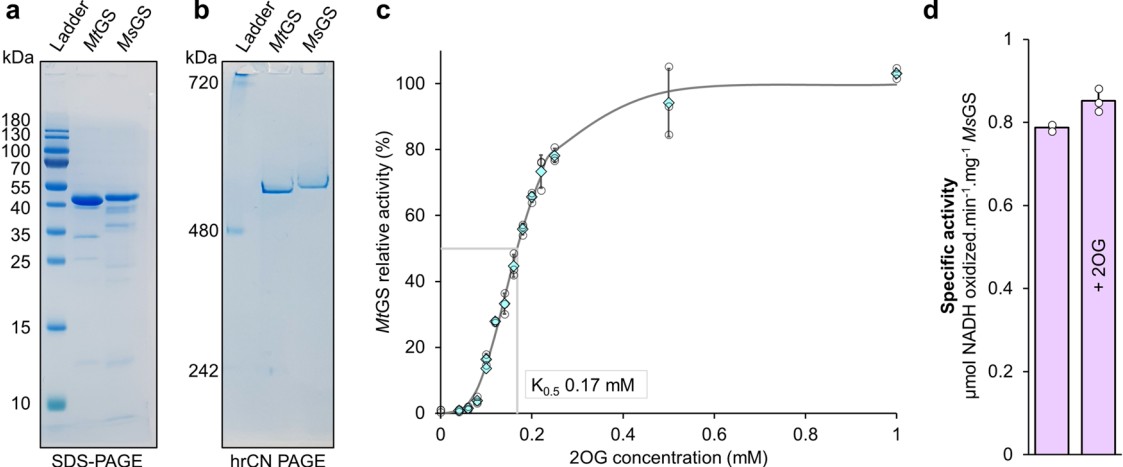

**Fig. 1 Purification and characterization of GS from methanogenic archaea. a** SDS-PAGE and (**b**) Native PAGE of 2 μg purified *Mt*GS and *Ms*GS. **c** Relative activity of *Mt*GS at different 2OG concentrations. The presented fit (grey plain line) is extrapolated from the determined Michaelis-Menten parameters. **d** The specific activity of *Ms*GS without or with 2 mM 2OG at 40 mM glutamate and 20 mM $NH_4Cl$. For **c** and **d** data is represented as mean (by cyan diamonds on panel (**c**)) ± standard deviation (s.d.) and individual values are shown as white circles ($n = 3$).

**Table 1 Kinetic parameters of *Mt*GS and *Ms*GS for $NH_4Cl$ and glutamate.**

| Enzyme | Substrate | App$K_m$ (mM) | App$V_{max}$ (U.mg$^{-1}$) | Hill coefficient |
|---|---|---|---|---|
| *Mt*GS | $NH_4Cl$ | 0.14 ± 0.01 | 3.269 ± 0.115 | 1.48 ± 0.21 |
| | Glutamate | 8.76 ± 0.87 | 3.419 ± 0.134 | 1.69 ± 0.23 |
| *Ms*GS | $NH_4Cl$ | 0.07 ± 0.01 | 0.98 ± 0.02 | 0.84 ± 0.12 |
| | Glutamate | 32.57 ± 3.13 | 1.40 ± 0.07 | 2.35 ± 0.41 |

All activities were measured in technical triplicates ($n = 3$). *Mt*GS was measured in the presence of 2 mM 2OG. The curves used for kinetics parameter determination are provided in Fig. S5.

methanogen *Methermicoccus shengliensis*. This archaeon belongs to the *Methanosarcinales* order and uses methylotrophic methanogenesis for energy and carbon acquisition[32]. Its genome codes for a single GSI-α (*Ms*GS) and no GDH. *Ms*GS was anaerobically purified from *M. shengliensis* leading to a major band at ~50 kDa on SDS PAGE (Fig. 1a). A dodecameric assembly was suggested by hrCN PAGE (estimated complex size at 547.5 kDa, Fig. 1b). Both the SDS and hrCN PAGE indicate a molecular weight for *Ms*GS (WP_042685700.1, predicted molecular weight: 49.53 kDa) superior to that of *Mt*GS (WP_018154487.1, predicted molecular weight: 50.24 kDa), a difference that we attributed to an artifact from the electrophoretic migration. When subjected to size exclusion chromatography, both proteins exhibited an elution volume in the range of the proposed dodecameric assembly in which *Ms*GS is smaller than *Mt*GS (Fig. S4). Native electrophoresis and size exclusion chromatography indicate a single oligomeric state for the purified GS. Unlike for *Mt*GS, 2OG was not required for *Ms*GS activity, and its addition (2 mM) did not affect its activity (Fig. 1d). Kinetic parameters of both enzymes were determined and are summarized in Table 1 (kinetic curves are provided in Fig. S5). Both enzymes exhibit the same order of magnitude for the apparent $K_m$ for glutamate and ammonium, and binding of both substrates exhibits a marginal positive cooperativity in *Mt*GS, while in *Ms*GS glutamate binding showed a positive cooperativity and almost no cooperativity for ammonium. Such observed differences in cooperativity between both enzymes might hide a more sophisticated discrepancy in their mode of regulation and sensing the intracellular metabolite balance.

Structural studies were undertaken to investigate the activation mechanism and the difference between both GS systems.

**Structural snapshots of archaeal GS**. The structure of *Mt*GS in its resting state was solved by using the diffraction data obtained by Engilberge et al.[31]. The data was collected on a crystal obtained by co-crystallization with a crystallophore (TbXo4) that corrected the twinning issue inherent to this crystalline form. The structure (*Mt*GS-apo-TbXo4) was solved by molecular replacement using the closest structural homolog from *Bacillus subtilis* (*Bs*GS resting state, PDB 4LNN)[18] and refined to a resolution of 1.65 Å (Table 2). The *Mt*GS structure presents the typical homo-dodecameric architecture as seen in other structural homologs, with a monomeric unit divided into N-ter (1-111) and C-ter (112-448) domains (Fig. 2a–c, Fig. S6). A root mean square deviation (rmsd) of 0.847 Å (on 338 Cα aligned from one monomer) exists between this structure and the apo structure of *Bs*GS (Fig. S7). The TbXo4, contributing to the crystal packing, might have artificially provoked this deviation. Moreover, terbium atoms are located in the active site, coordinated by the residues involved in the $Mg^{2+}/Mn^{2+}$ recognition (Fig. S8). To exclude any artefactual effect of TbXo4, we performed crystallization in the absence of the compound. The resulting crystal belonged to the same space group as the TbXo4-containing form, but the crystallographic data exhibited a pseudomerohedral twinning with a fraction of 0.12. The protein structure (*Mt*GS-apo-without TbXo4) was refined to 2.43 Å and is almost identical to the TbXo4-containing form (rmsd of 0.206 Å, 410 atoms aligned Fig. S8, residues 67-69 could not be modeled in the TbXo4-lacking structure), confirming that the observed conformation is not the artefactual result of the TbXo4 binding. For this reason, the *Mt*GS TbXo4-containing structure obtained at the higher resolution of 1.65 Å was used for further analyses.

The enzyme from *M. shengliensis* was crystallized in the absence of ligands, and two different crystalline forms, referred to as *Ms*GS-apo 1 and *Ms*GS-apo 2, were analyzed (Table 3). *Mt*GS-apo-TbXo4 was selected as a template for molecular replacement, and the structures were refined to 2.64 Å and 3.09 Å, respectively (Fig. S9). As *Mt*GS, the homolog from *M. shengliensis* has a dodecameric organization with the N-ter (1-106) and C-ter (107-442) domains forming the protomeric unit (Fig. 2d–f,

**Table 2 X-ray analysis statistics for *Mt*GS.**

|  | *Mt*GS-apo-TbXo4 | *Mt*GS-apo without TbXo4 | *Mt*GS- 2OG/Mg$^{2+}$/ATP | *Mt*GS-2OG/Mg$^{2+}$ |
|---|---|---|---|---|
| **Data collection** | | | | |
| Wavelength (Å) | 0.97625 | 1.00004 | 1.30511 | 0.99999 |
| Space group | C222$_1$ | C222$_1$ | P1 | P1 |
| Resolution (Å) | 49.78 – 1.65 | 76.67 - 2.43 | 110.03 - 2.15 | 203.54 - 2.91 |
|  | (1.74 – 1.65) | (2.57 - 2.43) | (2.30 - 2.15) | (3.12 - 2.91) |
| Cell dimensions | | | | |
| a, b, c (Å) | 131.43, 228.45, 204.80 | 132.65, 230.24, 205.59 | 112.34, 131.77, 131.51 | 131.81, 131.93, 203.54 |
| α, β, γ (°) | 90, 90, 90 | 90, 90, 90 | 60.04, 87.72, 67.34 | 89.95, 89.86, 60.05 |
| R$_{merge}$(%)$^a$ | 6.3 (91.8) | 26.9 (179.5) | 11.0 (95.0) | 12.2 (50.1) |
| R$_{pim}$ (%)$^a$ | 3.9 (56.6) | 7.3 (46.4) | 6.8 (59.0) | 7.6 (34.4) |
| CC$_{1/2}$ $^a$ | 0.999 (0.782) | 0.998 (0.782) | 0.995 (0.569) | 0.993 (0.548) |
| I/σ$_I$$^a$ | 11.9 (1.6) | 7.4 (1.5) | 9.3 (1.3) | 5.9 (1.6) |
| Spherical completeness$^a$ | 98.9 (97.9) | 86.6 (28.7) | 76.9 (21.1) | 64.8 (16.8) |
| Ellipsoidal completeness$^a$ | / | 93.3 (49.5) | 92.2 (58.8) | 91.7 (69.3) |
| Redundancy$^a$ | 6.9 (6.9) | 14.5 (15.6) | 3.6 (3.4) | 3.5 (3.0) |
| Nr. unique reflections$^a$ | 361,584 (51,999) | 102,291 (5,117) | 248,419 (12,420) | 169,224 (8,461) |
| **Refinement** | | | | |
| Resolution (Å) | 49.78 – 1.65 | 58.89 - 2.43 | 42.88 - 2.15 | 43.74 - 2.91 |
| Number of reflections | 361,148 | 102,286 | 248,376 | 169,113 |
| R$_{work}$/R$_{free}$$^b$ (%) | 16.30/18.50 | 22.94/26.82 | 17.10/20.20$^d$ | 25.41/27.80 |
| Number of atoms | | | | |
| Protein | 21,321 | 21,082 | 42,408 | 84,792 |
| Ligands/ions | 81 | 154 | 710 | 371 |
| Solvent | 2,932 | 293 | 3,041 | 0 |
| Mean B-value (Å$^2$) | 33.52 | 52.65 | 38.36 | 63.17 |
| Molprobity clash score | 2.32 | 4.01 | 1.99 | 1.09 |
| Ramachandran plot | | | | |
| Favored regions (%) | 98.20 | 96.51 | 97.92 | 96.26 |
| Outlier regions (%) | 0 | 0.04 | 0.22 | 0.24 |
| rmsd$^c$ bond lengths (Å) | 0.012 | 0.004 | 0.009 | 0.010 |
| rmsd$^c$ bond angles (°) | 1.485 | 0.654 | 0.93 | 1.274 |
| **PDB ID code** | 8OOL | 8OON | 8OOO | 8OOQ |

$^a$Values relative to the highest resolution shell are within parentheses.
$^b$R$_{free}$ was calculated as the R$_{work}$ for 5% of the reflections that were not included in the refinement.
$^c$rmsd, root mean square deviation.
$^d$R$_{work}$ and R$_{free}$ are from the PDB validation report.

Fig. S10). *Ms*GS-apo 1 was used for the following structural analyses due to the better resolution.

A more detailed view of the monomeric structures highlights the key catalytic elements surrounding the active site in both GS (Fig. 2c and f, Fig. S11[18]): the Glu-flap (*Mt*GS 306-311, *Ms*GS 299-304), the Tyr-loop (*Mt*GS 370-378, *Ms*GS 363-371), the Asn-loop (*Mt*GS 236-247, *Ms*GS 229-240), Tyr179-loop (*Mt*GS 153-163, *Ms*GS 146-156) and Asp50′-loop (*Mt*GS 57-71, *Ms*GS 51-65). A structural comparison of the monomeric unit reveals that *Ms*GS has a closer fit with *Bs*GS compared to *Mt*GS (rmsd *Ms*GS-*Bs*GS: 0.749 Å, 341 atoms aligned, *Mt*GS-*Bs*GS: 0.847 Å, 338 atoms aligned), which exhibits larger deviations such as an extended helix α2 resulting from a seven-residue insertion (Fig. 2g, Fig. S7). While some deviations of local loops might be attributed to the increase in flexibility (Fig. S7a), both archaeal GS deviate from *Bs*GS at the loop juxtaposed to the Tyr179-loop (*Mt*GS 164-171, *Ms*GS 157-164) before helix α6 (highlighted by a star on Fig. 2g, h) at the subunit interface.

The GS are well known to be dynamic, and structural rearrangement can occur upon ligand binding[6,18,19,33,34]. For instance, *Bs*GS undergoes dramatic intersubunit conformational movements between the apo and transition state. Therefore, to investigate why *Mt*GS apo resting state would be catalytically inactive, co-crystallization with 2OG was undergone.

**A 2OG allosteric site localized at the inter-subunit junction.**
The complex with 2OG/Mg$^{2+}$ was obtained with and without

ATP (*Mt*GS-2OG/Mg$^{2+}$/ATP, *Mt*GS-2OG/Mg$^{2+}$) in a new crystalline form and both were refined to 2.15 Å and 2.91 Å resolution, respectively (Figs. 3 and 4, Table 2). Both 2OG-containing structures show high tridimensional conservation (rmsd 0.279 Å, 441 atoms aligned, Fig. S12) while deviating from the apo resting state (2OG/Mg$^{2+}$/ATP:rmsd 0.694 Å, 374 atoms aligned, 2OG/Mg$^{2+}$: 0.631 Å, 371 atoms aligned, Fig. S13 and S14). Since the 2OG/Mg$^{2+}$/ATP complex is at a higher resolution, this structure was preferred for deeper structural analyses (Fig. S15). The allosteric 2OG binding site, ~15 Å distant from the catalytic cleft, is localized at the interface between the C-terminal domain and the adjacent N-terminal domain. 2OG is coordinated via ionic bonds by Arg20′, Arg88′, Arg174, Arg175 and a hydrogen bond via Ser191 (primed numbers indicate the adjacent unit, Fig. 3a). Phe18′ stabilizes the 2OG via stacking with its phenyl ring. In the apo structure, water molecules replace 2OG in a pocket with a similar volume and slightly open to the solvent (Fig. S15c). In the 2OG-insensitive *Ms*GS structure, the side chain of Glu89′ substituting the *Mt*GS Val94′ would collide with the 2OG position, and its negative charge would additionally lead to the ionic repulsion of 2OG (Fig. 3b). Moreover, in *Ms*GS, most of the residues relevant for 2OG coordination (Arg88′, Arg174 and Ser191) are substituted. Similarly, the structures of the bacterial homologs show a tyrosine and glutamate at the equivalent positions 18′ and 34′ in *Mt*GS that would collide with the modeled 2OG and would contribute to ionic repulsion (Fig. 3c). Additionally, the 2OG-coordinating Arg88′ is absent (Fig. 3d). The

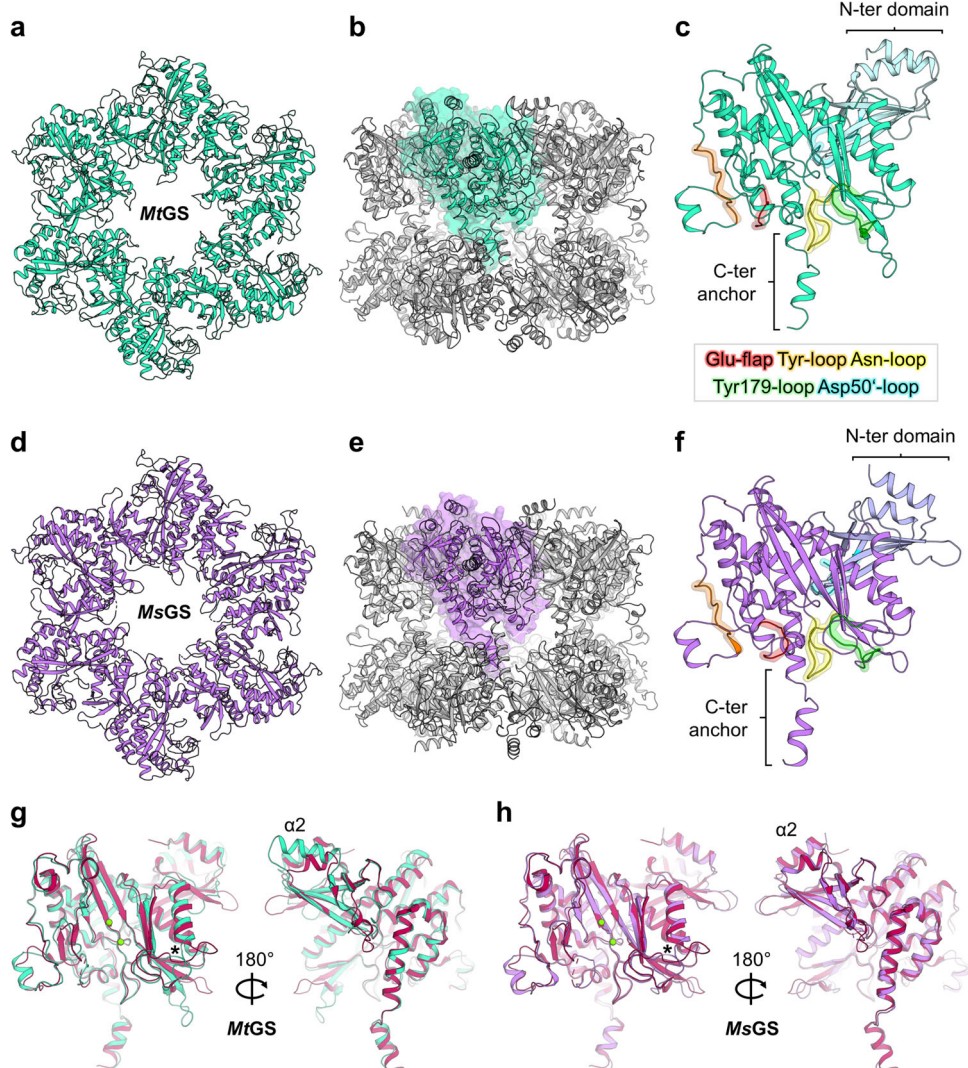

**Fig. 2 Structural organization of archaeal GS.** All models are in the apo state and represented in cartoons. **a** Top view of one *Mt*GS hexamer. **b** Side view of the *Mt*GS dodecamer with one subunit shown in a transparent surface (cyan). **c** *Mt*GS monomer with the main loops highlighted. The color coding of the loops is indicated in the box. The N- and C- terminal domains are colored light blue and cyan, respectively. **d** Top view of one *Ms*GS hexamer. **e** The side view of the *Ms*GS dodecamer with one subunit shown as a transparent surface (purple). **f** *Ms*GS monomer with the main loops highlighted with the same color coding as in panel (**c**). The N- and C-terminal domains are colored lavender and purple, respectively. **g** Overlay of *Mt*GS (cyan) and *Bs*GS apo state (red, PDB 4LNN). **h** Overlay of *Ms*GS (purple) and *Bs*GS apo state (red). For panels (**g**, **h**) the star indicates the position of the loop deviating in both archaeal GS compared to *Bs*GS. Mg atoms are displayed as green spheres.

Arg88′ might be the essential component for 2OG-based activation as it is positioned on the loop that undergoes the largest conformational change during 2OG binding. Therefore, its movement provoked by 2OG binding would trigger further motions throughout the polypeptide chain (see Supplementary Movie).

**2OG-dependent motions and active site remodeling in *Mt*GS.** 2OG presence at the interface of monomers reorients the bulky side chain of Phe18', and remodels salt bridge networks in its surrounding. Consequently, the N-terminal domain is pushed away from the neighboring C-terminal domain, and the α-helix 6 and β-hairpin 9-10 (residues 167-206, in the vicinity of the active site) are shifted (Fig. 3e). The structural alignment on the C-terminal part from one monomer exemplifies a 4.8° oscillation of the N-terminal domain, leading to a movement of 3 Å (Fig. S13b). The repositioning of the N-terminal domain would

clash with the next monomer, and therefore a domino effect occurs in the hexamer ring upon 2OG binding. This probably explains the cooperative behavior observed in the presented kinetic data (Hill coefficient 3.4 for 2OG, Fig. 1c). Overall, the transition between *Mt*GS resting and 2OG-bound states provokes a compaction of the outer ring and an opening in the center and middle parts (Fig. 3f), leading to a more "opened" conformation. The observed switch and the 2OG-bound active state are similar to the one observed between apo and transition state complex of *B. subtilis* (Fig. S16). The most dramatic change occurs in the Asp50′-loop (critical for glutamate and ammonia coordination) interacting with the loop 318-325, which reorganize to reach a position similar to that observed in the bacterial homologs co-crystallized with the L-methionine-S-sulfoximine-phosphate (abbreviated as SOX, a mimic of a reaction transition state)[18,19] (Fig. 5a–c and S17).

In the *Mt*GS-2OG/Mg$^{2+}$/ATP structure, the ATP is located in a similar position compared to bacterial homologs (Fig. 4a, b).

**Table 3 X-ray analysis statistics for *Ms*GS.**

| | *Ms*GS-apo 1 | *Ms*GS-apo 2 | *Ms*GS- Mg$^{2+}$/ATP |
|---|---|---|---|
| **Data collection** | | | |
| Wavelength (Å) | 1.00004 | 1.00002 | 0.97949 |
| Space group | $P2_1$ | $P4_332$ | $P2_1$ |
| Resolution (Å) | 109.99 - 2.64 | 130.75 - 3.09 | 78.85 - 2.70 |
| | (2.93 - 2.64) | (3.17 - 3.09) | (2.88 - 2.70) |
| Cell dimensions | | | |
| a, b, c (Å) | 130.92, 195.65, 133.44 | 226.46, 226.46, 226.46 | 131.55, 197.40, 135.17 |
| α, β, γ (°) | 90, 94.71, 90 | 90, 90, 90 | 90, 94.89, 90 |
| $R_{merge}$(%)[a] | 7.6 (87.0) | 34.5 (309.9) | 6.2 (68.8) |
| $R_{pim}$ (%)[a] | 3.5 (46.7) | 6.8 (60.7) | 3.3 (37.0) |
| $CC_{1/2}$ [a] | 0.999 (0.600) | 0.998 (0.486) | 0.999 (0.701) |
| $I/\sigma_I$[a] | 15.5 (1.6) | 12.2 (1.4) | 17.4 (2.0) |
| Spherical completeness[a] | 62.8 (11.5) | 96.6 (63.9) | 68.5 (19.8) |
| Ellipsoidal completeness[a] | 90.3 (78.7) | 96.6 (63.8) | 88.1 (89.8) |
| Redundancy[a] | 5.7 (4.4) | 26.6 (26.8) | 4.3 (4.4) |
| Nr. unique reflections[a] | 123,574 (6,180) | 35,711 (1,793) | 128,572 (6,429) |
| **Refinement** | | | |
| Resolution (Å) | 62.96 – 2.64 | 65.37 – 3.09 | 49.10 – 2.70 |
| Number of reflections | 123,529 | 35,696 | 128,533 |
| $R_{work}/R_{free}$[b] (%) | 19.14/22.52 | 18.63/21.62 | 19.61/21.72 |
| Number of atoms | | | |
| Protein | 40,821 | 6,920 | 40,815 |
| Ligands/ions | 76 | 64 | 431 |
| Solvent | 66 | 0 | 33 |
| Mean B-value (Å$^2$) | 72.24 | 80.11 | 70.78 |
| Molprobity clash score | 2.19 | 2.30 | 2.60 |
| Ramachandran plot | | | |
| Favored regions (%) | 98.14 | 96.31 | 97.95 |
| Outlier regions (%) | 0 | 0 | 0.02 |
| rmsd[c] bond lengths (Å) | 0.011 | 0.004 | 0.011 |
| rmsd[c] bond angles (°) | 1.35 | 0.678 | 1.348 |
| **PDB ID code** | 8OOW | 8OOX | 8OOZ |

[a]Values relative to the highest resolution shell are within parentheses.
[b]$R_{free}$ was calculated as the $R_{work}$ for 5% of the reflections that were not included in the refinement.
[c]rmsd, root mean square deviation.

The adenosine part is stacked in between Phe206 and Arg336 and coordinated via hydrogen bonds with Lys333 main chain and Ser254 side chain. The ribose moiety is coordinated by the Phe204 and Phe206 main chains and the Glu189 side chain. His252, Arg321, Arg326, and Arg336 bind the triphosphate backbone with ionic bonds. In the resting state, the ATP would clash with the Phe206 and would not bind without displacing the β-sheet 10 (residues 199-205) and the following loop (206-209) (Fig. 4c and S18).

We also obtained and refined to 2.70 Å a *Ms*GS structure in complex with ATP and Mg$^{2+}$ (Table 3). Only minor differences exist compared to the apo state (Fig. S14d). While the adenine part of the ATP is bound in a similar way as described before, the ribose backbone is tilted by 90 °, pointing the triphosphate away from the active site (Fig. 4d). This results in a non-catalytic state, similar to the one observed in the *Bs*GS structure loaded with the non-hydrolysable ATP analogue AMPPCP (PDB 4LNK)[18].

Nevertheless, a sequence alignment of *Bs*GS, *Mt*GS and *Ms*GS revealed a complete conservation of the residues involved in nucleotide binding (Fig. 4e). Therefore, we concluded that the three GS share the same ATP coordination throughout the catalytic steps.

2OG binding in *Mt*GS also impacts the glutamate binding site architecture. The residues involved in glutamate binding (or the mimic SOX molecule) in *B. subtilis* are perfectly conserved in *Mt*GS, *Ms*GS and other archaeal GS (Fig. 5). The conservation between the archaeal and bacterial domains reinforces the importance of these residues to orchestrate the catalysis. While the predicted residues involved in Mg$^{2+}$/glutamate binding share the same position between the *Mt*GS resting and 2OG-bound states (Glu137, Glu139, Glu194, Gly246, His250, Glu309, Arg303, Arg340), the catalytic Arg321 (homologous to Arg316 in *Bs*GS) shows a drastic difference of position (Fig. 5b, c). In the resting state, the loop 318-325 is constrained by a network of hydrogen bonds and salt bridges from the Asp50′-loop, the side chain of the Arg321 itself being sequestered by Glu70′. Strikingly, in the 2OG-bound state of *Mt*GS, the 318-325 and Asp50′-loops adopt a position similar to all described active transition states in bacterial homologs (Fig. S17). This suggests that the binding of 2OG triggers conformational changes leading to a catalytically competent conformation and explaining the 2OG dependency for activity. Moreover, the Glu309 from the Glu-flap is disengaged and in a more relaxed state in the 2OG-bound model, allowing its assistance for catalysis. In *Ms*GS, the β-hairpin β3-β4 carrying the Asp50′-loop appears to adopt a similar position with or without ATP. However, the Asp50′ loop is flexible and cannot be modeled. The flexibility might directly affect the stabilization of the adjacent loop 313-319 (318-325 in *Mt*GS) that exhibits a different position than that found in *Mt*GS and bacterial GS with the catalytically important Arg314 (Arg321 in *Mt*GS) retracted on the Tyr-loop (Fig. 5d).

**MsGS is feedback inhibited by glutamine.** *Bs*GS is inhibited by the reaction product glutamine[18,19], a feedback regulation under the control of the Asp50′ loop. When glutamine occupies the substrate-binding site, it interacts with Glu304 (*Bs*GS numbering), required for catalysis. The Arg62′ from the Asp50′ loop stabilizes this interaction, resulting in a lock of the Glu-flap, closing the active site. The Arg62′ is not conserved in *Mt*GS (substituted by Gly67′) but is present in *Ms*GS (Arg61′, Fig. 5e). To investigate this feedback inhibition mechanism, we performed activity assays with glutamine addition for *Mt*GS and *Ms*GS. As suspected from the sequences, the addition of glutamine (20 mM) inhibited the activity of *Ms*GS by 97%, while no difference was observed for *Mt*GS (Fig. 5f). The substitution of the arginine appears to completely abolish glutamine inhibition.

**Direct regulation by 2OG and glutamine within the domain Archaea.** In addition to the previous studies of archaeal GSI-α[26–28], the characterization of the enzyme from *M. thermolithotrophicus* and *M. shengliensis* highlighted that the mechanisms of post-translational regulation are not conserved among archaea. Our study points out that the direct binding of 2OG and glutamine depends on a restricted number of residues, suggesting that the potential regulation of GS activity by both 2OG and glutamine might be predictable from the amino acid sequences. The conservation of residues involved in regulator-binding was analyzed through 500 GS sequences regrouping *Mt*GS and its closest archaeal homologs in the RefSeq database (Fig. 6 and Fig. S19). The GSI-α sequences used to build the tree are, in most cases, forming monophyletic branches that separate archaeal orders (sometimes different groups of GS within orders)

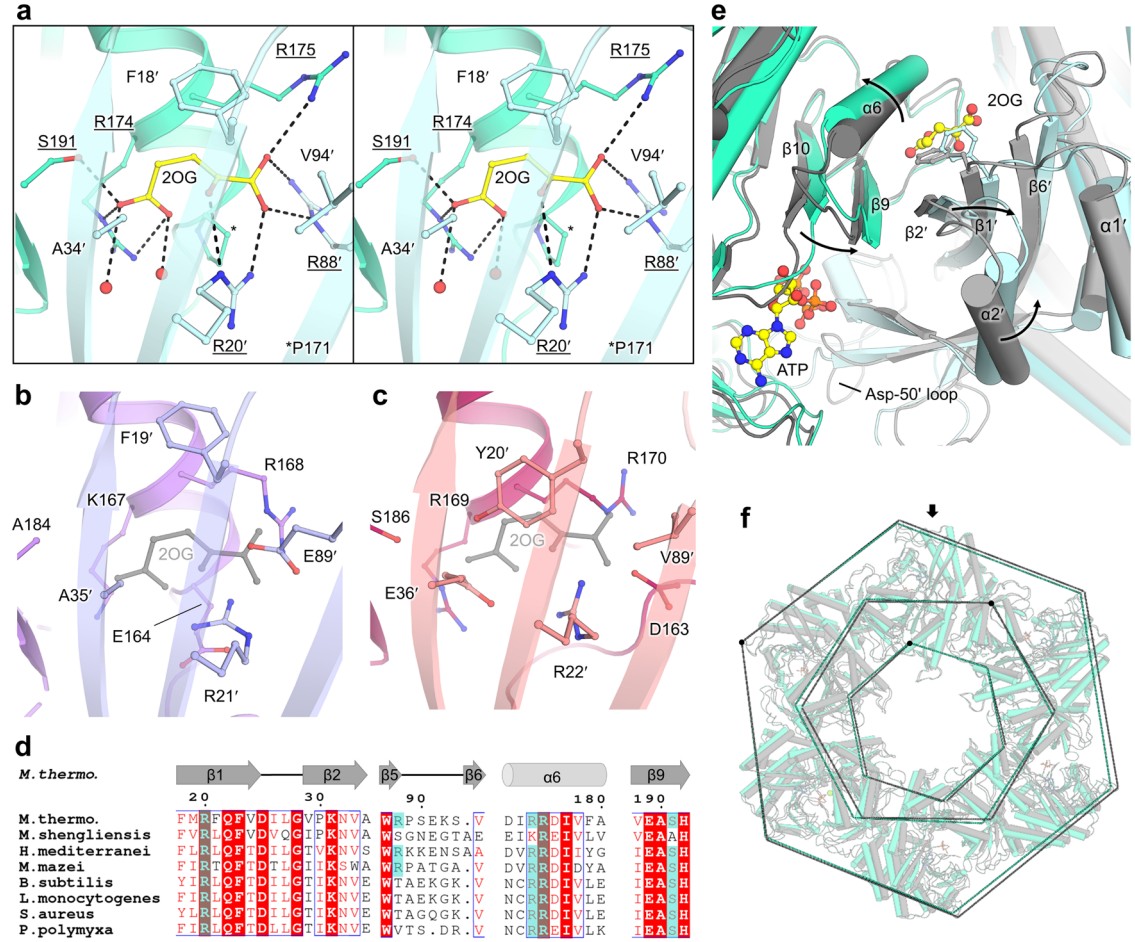

**Fig. 3 2OG binding site and structural rearrangement in *Mt*GS. a** Close-up of the 2OG binding site in *Mt*GS (cyan cartoon, the adjacent monomer in light blue) shown as a stereo view. 2OG and the residues in its vicinity are shown as balls and sticks with contacts in black dashes. **b**, **c** Same view as in (**a**) showing *Ms*GS apo ((**b**) purple cartoon, with the adjacent subunit in light purple) and *Bs*GS apo ((**c**) red cartoon, with the adjacent subunit in light red, PDB 4LNN). 2OG from *Mt*GS (gray) was superposed to visualize the clash with E89′ for *Ms*GS and E36′/Y20′ for *Bs*GS. **d** Sequence alignment of different GSI-α in which 2OG-binding residues observed in *Mt*GS are highlighted with a cyan box (see Fig. S11 for the entire alignment). **e** Structural rearrangements between the apo (gray cartoon) and 2OG/Mg$^{2+}$/ATP bound state (cyan cartoon). The adjacent monomer is colored lighter. Phe18′ is shown as sticks. Arrows highlight the movements caused by 2OG binding. **f** *Mt*GS apo (gray cartoon) superposed to *Mt*GS-2OG/Mg$^{2+}$/ATP (cyan cartoon). The superposition was done on one monomer (indicated by an arrow), and a dashed line was drawn on the Cα position of Val4, Gly198, and Asn267 to illustrate the overall movements. For all, oxygen, nitrogen, and phosphorus are colored in red, blue, and orange, respectively. Carbons are colored depending on the chain and in yellow for ligands.

and that apparently share a regulation mechanism (see residue conservation in Fig. S19), except for some sequences including *Ms*GS which do not clearly branch with any other GS (Fig. 6). Around half of the GS groups isolated from this tree (8 over the 17 archaeal GS groups) appear to harbor the residues necessary for binding either 2OG or glutamine. In comparison, a third of the GS groups can be theoretically regulated by both (6 over the 17 archaeal GS groups, Fig. 6), of which the enzymes from *H. mediterranei* and *M. mazei* are representatives. In certain archaeal orders (e.g. *Methanosarcinales*), the genome encodes two GS isoforms that appear to have different sensitivity toward 2OG and glutamine. The enzymes found in *Thermococcales*, *Thermoplasmatales* or *Methanomassiliicoccales* present substitutions that would hinder their capacities to bind both 2OG and glutamine. Yet, binding at another position in the protein structure cannot be excluded to allow potential interactions with regulatory partners. This analysis, yet restricted to a small number of archaeal orders and therefore far from describing the overall archaeal GS family, suggests variability in regulation strategies among these enzymes pointed out by the enzymatic and structural analyses.

## Discussion

In the absence of a GDH system, GS represents the main entry point for ammonia assimilation, feeding the cellular metabolism with nitrogen. The ATP-dependent reaction must be under tight control to fit cellular needs. The GS, as does their regulation, come in different flavors in eukaryotes and prokaryotes[8,18,19,26–28]. Many studies have been performed on bacterial and eukaryotic enzymes, but only a few are focused on the GS from the domain *Archaea*. This work unveiled the first GSI-α crystal structures. Their homododecameric organization is highly similar to their bacterial counterparts. This is in accordance with published phylogenetic analyses in which archaeal and bacterial GSI-α are part of a monophyletic group (Fig. 6)[9]. Our results expand the knowledge gathered on this enzyme and aim to dissect the regulation mechanism at the molecular level.

2OG is a sensor for cellular nitrogen availability and has recently been recognized as a master regulator of multiple biochemical pathways[30], especially other elements of nitrogen assimilation. For instance, most of the P$_{II}$-family regulatory proteins, operating the signaling for nitrogen fluxes (including the

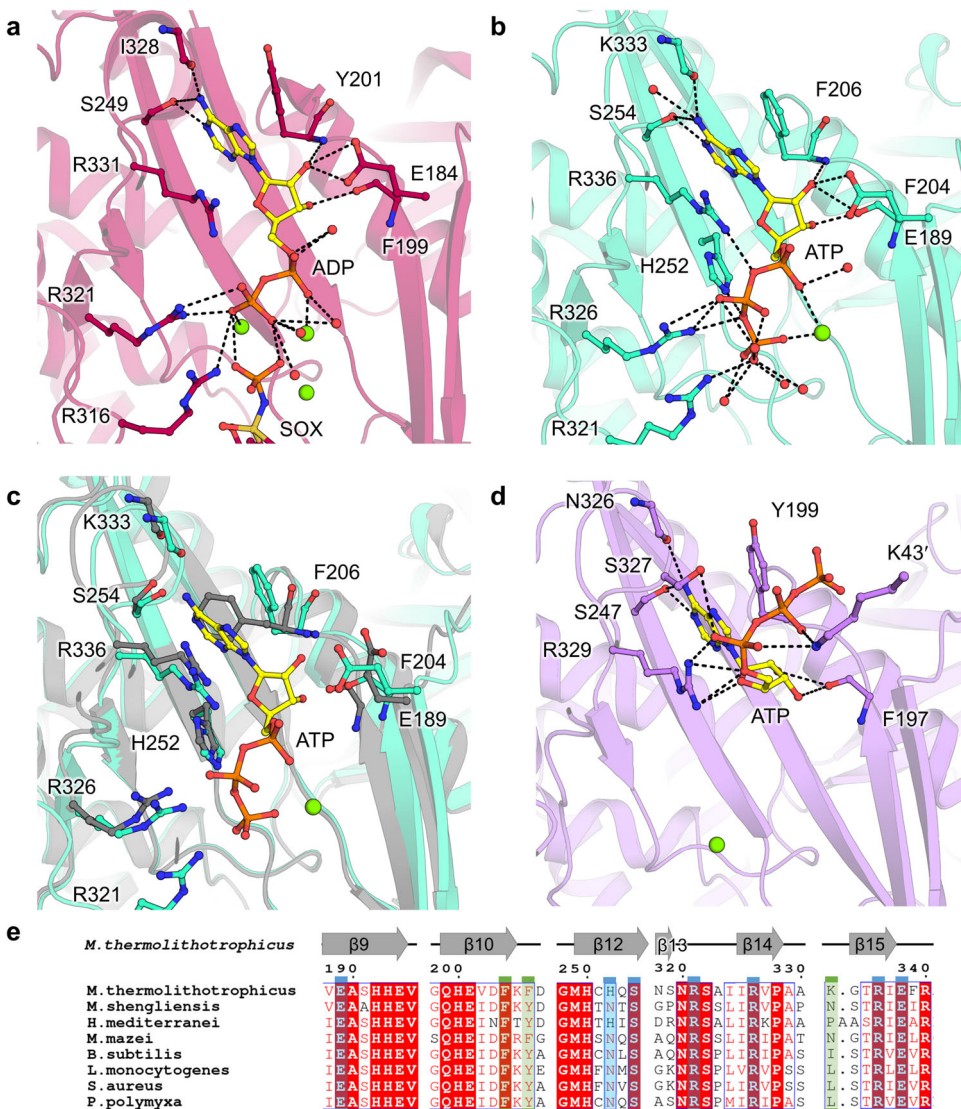

**Fig. 4 ATP-binding site comparison between different GSI-α. a** ATP binding site in *Bs*GS transition state (containing SOX/Mg$^{2+}$/ADP, PDB 4LNI) and (**b**) *Mt*GS-2OG/Mg$^{2+}$/ATP. **c** Superposition of the C-terminal domain of *Mt*GS apo (gray) on *Mt*GS-2OG/Mg$^{2+}$/ATP (cyan), with an overlay of the ATP-binding residues. **d** ATP binding site in *Ms*GS-Mg$^{2+}$/ATP. Models are represented in transparent cartoons with the ligands (yellow) and interacting residues shown as balls and sticks. Oxygen, nitrogen, sulfur, phosphorus, and magnesium are colored red, blue, dark yellow, orange, and green, respectively. Carbons are colored by chain and ATP carbons in yellow. Hydrogen bonds are visualized as black dashes. **e** Sequence alignment of the ATP binding residues. Residues coordinating the nucleotide via side chain and main chain hydrogen bonds are highlighted by a blue and green box, respectively (based on *Mt*GS). *Ms*GS K43'/S327 were omitted from the analysis due to the artefactual γ-phosphate position.

regulation of GS), are regulated by 2OG[5,30,35–37]. Previous studies highlighted 2OG as a direct regulator of GSI-α in *Archaea*[26–28], such as *Haloferax mediteraneii* and *Methanosarcina mazei*, in which 2OG activates the enzyme by 12-fold[26] and 16-fold[28], respectively. Our study found that *Mt*GS activity is strictly dependent on this metabolite under the described experimental conditions. *Mt*GS activation is saturated at 0.6 mM 2OG and is inactive below 0.06 mM. This range remarkably fits the measured 2OG cellular concentration in the methanogen *Methanococcus maripaludis*, which is around 0.8 mM under N$_2$-fixing conditions (nitrogen-limited) and 0.08 mM after ammonium addition[38]. Relying on 2OG concentration as a sensor of cellular nitrogen availability could benefit an energy-limited organism such as *M. thermolithotrophicus*, and might represent a "primitive" regulation mechanism, which evolved before more elaborate regulatory networks emerged. In addition, regulation at the transcriptional level also occurs as *M. thermolithotrophicus* upregulates *glnA*

expression when cells switch to N$_2$ fixation[39], and it is known that small RNAs are involved in nitrogen metabolism regulation in prokaryotes, including methanogens[40].

The structure of *Mt*GS reveals an allosteric pocket perfectly suited to accommodate 2OG via specific salt bridges, hydrogen bonds, and Van der Waals contacts. Binding of 2OG leads to a succession of conformational rearrangements resulting in a catalytically competent state. A conservation of all five residues of the 2OG binding site can be found in other orders of archaea (Fig. 6 and S19). However, while the motif would allow 2OG binding, the influence on the activity remains to be verified as several other structural features are involved in the accurate positioning of catalytic residues. A 2OG binding motif is far from being a feature shared by all archaea, as exemplified by the characterization of the 2OG-insensitive *Ms*GS. Previous works on the enzymes from *H. mediterranei* and *M. mazei* pointed towards a unified 2OG activation mechanism as main regulation in

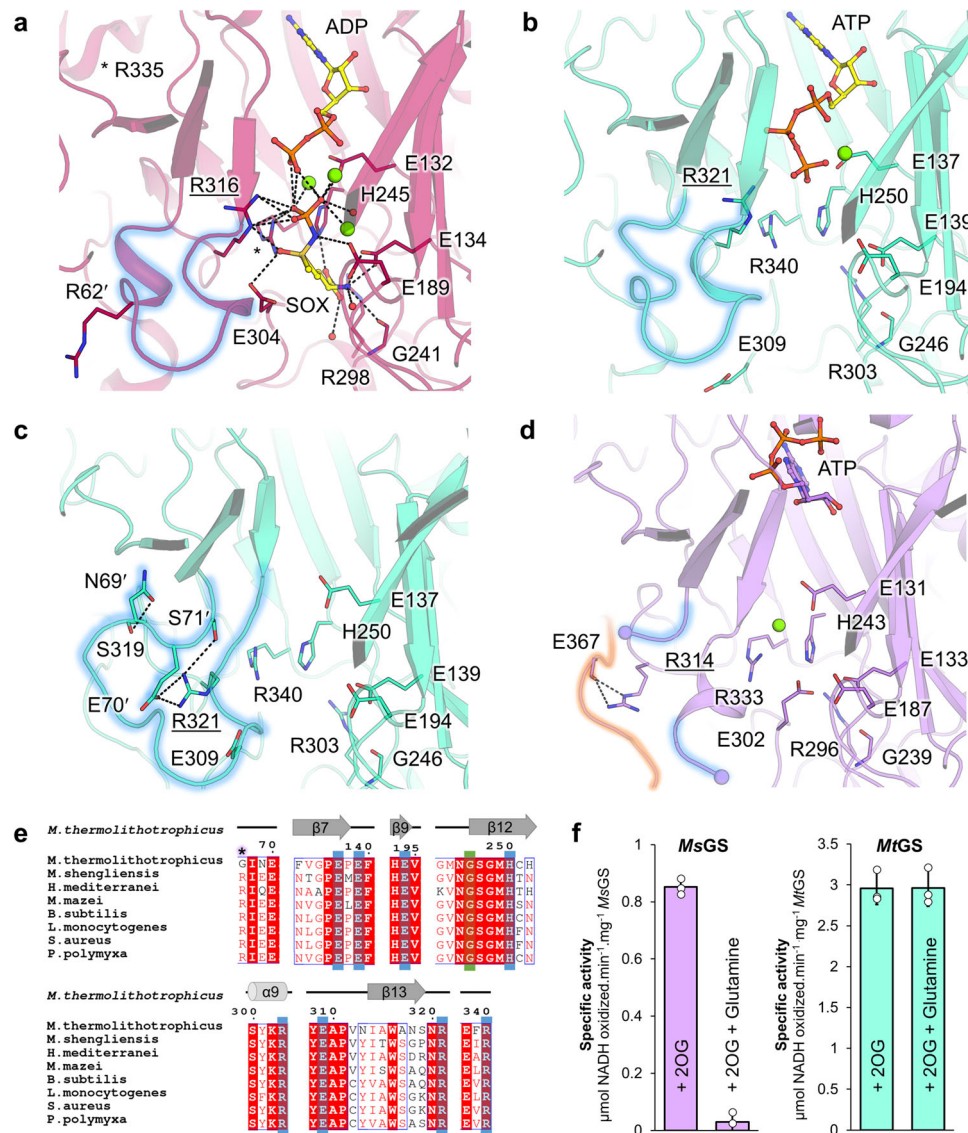

**Fig. 5 Glutamate-binding site comparison and glutamine feedback inhibition. a–d** Glutamate binding site in **a** BsGS-SOX/Mg$^{2+}$/ADP (red, 4LNI), (**b**) MtGS-2OG/Mg$^{2+}$/ATP (cyan), (**c**) MtGS apo (cyan), and (**d**) MsGS-Mg$^{2+}$/ATP (purple). Models are in cartoons with ligands and equivalent residues binding SOX as balls and sticks. Oxygen, nitrogen, sulfur, phosphorus, and magnesium are colored red, blue, dark yellow, orange, and green, respectively. Carbons are colored by chain and ATP carbons in yellow. Hydrogen bonds are visualized as black dashes. The Asp-50′ loop region is highlighted by a blue glow. For MsGS, the Tyr-loop is highlighted with an orange glow. An underlined label highlights the catalytic arginine (e.g., R321 in MtGS). **e** Alignment of the residues involved in glutamate binding (based on BsGS). Side chain and main chain interactions are highlighted by a blue and green box, respectively. A star highlights the arginine responsible for glutamine feedback inhibition in BsGS. **f** Specific activity in the absence and presence of glutamine in both archaeal GS. Data is represented as mean ± s.d and individual values are shown as white circles (n = 3).

archaeal GS. However, spreading the analysis to other species rather highlights a variety of different regulation mechanisms depending on archaeal groups and enzyme isoforms (Fig. 6). The glutamine inhibition for MsGS, previously described in BsGS[18], is dependent on one essential arginine (R62′) responsible for the Glu-flap sequestration preventing the release of the product from the active site, which is not happening in MtGS due to a substitution (Gly67′).

Even if both, 2OG and glutamine, act through different mechanisms, e.g. competitive inhibition versus allosteric activation, it is interesting that their modulation acts on the same key determinants. In the resting state of MtGS without 2OG or the glutamine feedback-inhibited BsGS, the Glu-flap and the catalytic arginine are locked in an unproductive conformation by the Asp50′-loop. In MtGS the 2OG binding provokes a major

displacement of the N-terminal domain towards the adjacent C-terminal domain and the overall conformational changes promote a restructuration of the Asp50′-loop, leading to the repositioning of Arg321 and the Glu-flap, yielding an active conformation. Additionally, the 2OG-dependent motions provide enough flexibility to allow the correct structuration of the Mg$^{2+}$/ATP binding site, similar to MsGS in its resting state.

Kinetic measurements revealed similar apparent $K_m$ for ammonia and glutamate between both GS (Table 1) and are comparable to previous studies[18,41]. The determined $V_{max}$ also lies within the previously reported range of specific activity for archaeal and bacterial GS (up to around 6 μmol.min$^{-1}$.mg$^{-1}$ in both H. mediterranei and M. mazei[26,28] and up to 23 μmol.min$^{-1}$.mg$^{-1}$ in B. subtilis[42]). P$_{II}$-family proteins might play a role in MsGS and MtGS activation that could mediate 2OG

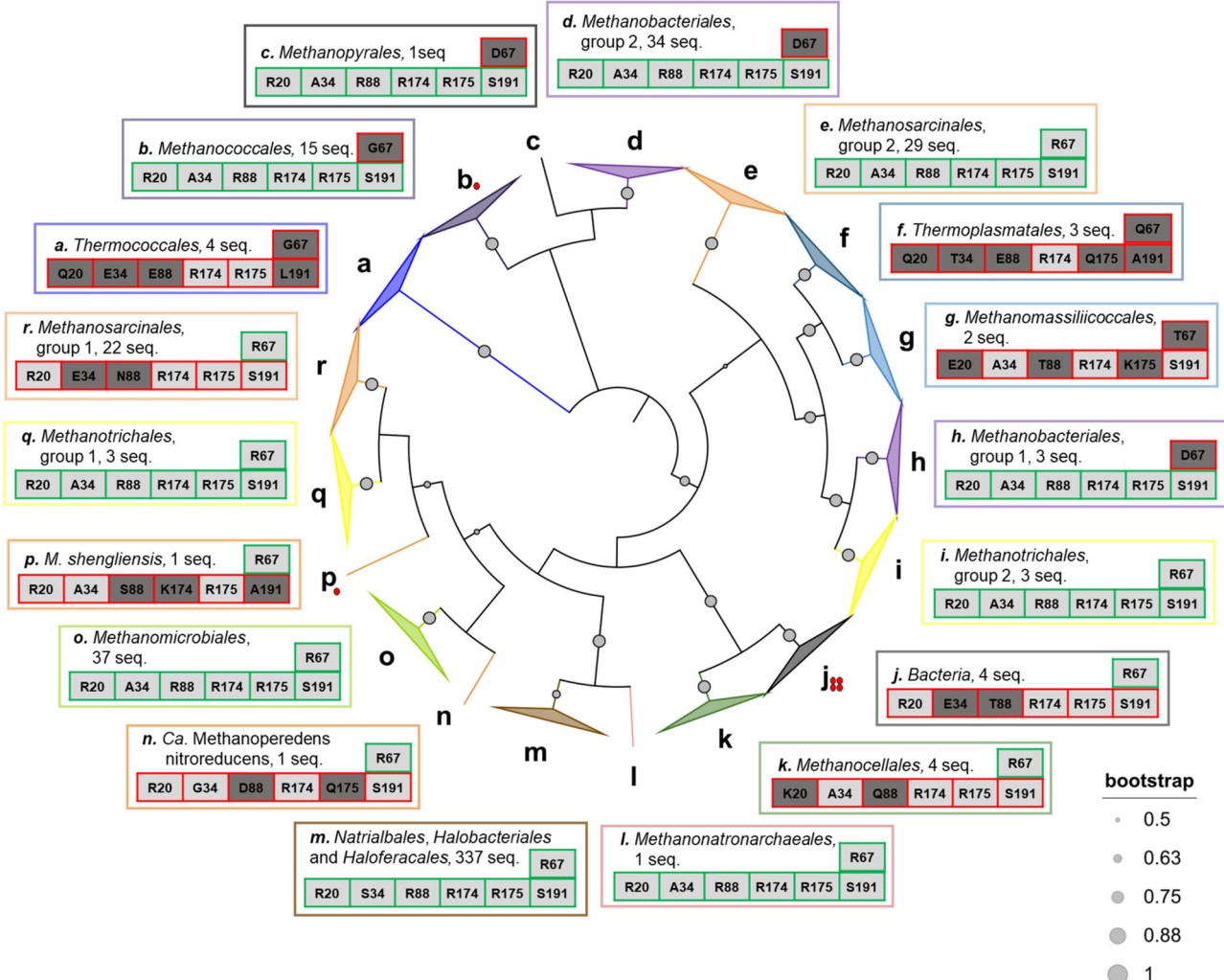

**Fig. 6 Conservation of residues binding 2OG and glutamine in archaeal GS.** The presented phylogenic inner tree (maximum likelihood) was constructed with the 500 closest sequences to *Mt*GS in the RefSeq database, restricted to the domain *Archaea*, as well as the sequences of the bacterial GSI-α from *B. subtilis*, *S. aureus*, *L. monocytogenes* and *P. polymyxa*. The tree was colored by orders (except for *Bacteria*), and gray dots with different radii represent the bootstrap support of each branch. Branches containing monophyletic groups are collapsed. Sequences forming monophyletic branches are gathered and labeled by a letter. The branch containing 337 sequences belonging to *Natrialbales*, *Halobacteriales*, and *Haloferacales* orders is condensed as no clear monophyletic groups could have been extracted. The outer panels display the most common residues at equivalent positions involved in 2OG (Arg20, Arg88, Arg174, Arg175, and Ser191, *Mt*GS numbering, bottom line) and glutamine (position 67 in *Mt*GS, upper line) binding. Panels are framed and labeled with the color and letter used in the inner tree. Ala34 in *Mt*GS is not involved in 2OG coordination, but a substitution by a bulky residue (e.g. glutamate) would hinder its fixation. The residues are colored in light or dark gray depending on whether they allow metabolite binding or not, respectively. The residue distribution for each position in each group is presented in Fig. S19, and the sequences can be found in Supplementary Data 2. The GS that are predicted to be able or unable to bind 2OG and glutamine are framed in green and red, respectively. Red dots indicate the GSI-α structurally characterized previously or in the present work.

sensitivity, a hypothesis that future exploratory studies will hopefully confirm. In contrast, the sP26 protein described in *M. mazei* is not encoded in the genomes of *M. thermolithotrophicus* and *M. shengliensis*, and the same applies to the repressor GlnR forming higher oligomer species in the bacterial systems. However, non-homologous functional equivalents may exist. Similarly, if our experiments do not suggest any modification of the oligomeric state, such a regulatory mechanism could occur in the cell in the presence of other regulatory partners, which might become separated during the purification process.

The works gathered on GS illustrate that despite the universal requirement of assimilating ammonia to fuel nitrogen metabolism and a conserved reaction mechanism, microbes rely on different ways to modulate this ATP-dependent activity. Such regulatory networks might have been elaborated over time to adapt to a particular physiology, environment, or catabolism. Opening scientific investigation to a broader group of organisms will hopefully contribute to extending our knowledge of this crucial and ancient enzyme.

## Materials and methods

**Cultivation**. *M. thermolithotrophicus* (DSM 2095) cells were obtained from the Leibniz Institute DSMZ - German Collection of Microorganisms and Cell Cultures (Braunschweig, Germany). Cells were grown in a mineral medium as described in Jespersen et al.[43]. Anaerobic cultivation of *M. thermolithotrophicus* was performed in a fermenter with $NH_4Cl$ and sulfate as described in Jespersen & Wagner[44] with slight modifications, or in a fermenter with $N_2$ and sulfate as described in Maslać et al.[39]. We did not observe dramatic changes in the

expression of GS nor its properties between both cultivation processes. *M. shengliensis* ZC-1 (DSM 18856) was also obtained from the DSMZ (Braunschweig, Germany) and was grown anaerobically on methanol as previously described in Kurth and Müller et al.[45].

**Purification**. Cell lysis and extracts were prepared in an anaerobic chamber at room temperature filled with an $N_2/CO_2$ atmosphere (90:10%). Enzyme purification was carried out under anaerobic conditions in a Coy tent filled with an $N_2/H_2$ atmosphere (97:3%), at 20 °C and under yellow light. For each step, chromatography columns were washed with at least three column volumes (CV) of the corresponding loading buffer, and samples were filtrated on 0.2 μm filters (Sartorius, Germany) prior to loading. During purification, the enzyme was followed by sodium dodecyl sulfate polyacrylamide gel electrophoresis (SDS-PAGE) and absorbance monitoring at 280 nm. Purifications were performed with adapting protocols, and the optimal one is described below for *Mt*GS and *Ms*GS.

*Mt*GS *purification*. 25 g (wet weight) cells were thawed and diluted in 200 ml lysis buffer (50 mM Tricine/NaOH pH 8.0, 2 mM dithiothreitol (DTT)), sonicated (~5 × 10 s at 60%, probe KE76 Bandelin SONOPULS, Germany) and centrifuged at 45,000 x *g* for 45 min at 18 °C. The supernatant was passed twice on a 4 × 5 ml HiTrap[TM] DEAE Sepharose FF column (Cytiva, Sweden). Elution was performed with a NaCl gradient ranging from 150 to 500 mM in the same buffer. The gradient was applied for 90 min at 2 ml.min$^{-1}$, and *Mt*GS was eluted at 170 to 260 mM NaCl. The pool was diluted 1:4 in 20 mM $NaH_2PO_4$ pH 7.6, 2 mM DTT. The sample was passed twice on 2 x Mini CHT [TM] type I column (Bio-Rad, United States). Elution was performed with a gradient of 20 to 200 mM $NaH_2PO_4$ in 10 min, followed by a second gradient from 200 to 500 mM $NaH_2PO_4$ in 10 min. The gradient was run at 1.5 ml.min$^{-1}$. Under these conditions, *Mt*GS was eluted at 20 to 200 mM $NaH_2PO_4$. The resulting pool was concentrated to 900 μl with a 30-kDa cutoff concentrator (Sartorius, Germany). The sample was injected thrice on a Superose[TM] 6 Increase 10/300 GL column (Cytiva, Sweden). The protein was eluted at 0.4 ml.min$^{-1}$ in 25 mM Tris/HCl pH 7.6, 10% glycerol, 150 mM NaCl, 2 mM DTT. The resulting pool was diluted 1:1 in 25 mM Tris/HCl pH 7.6, 2 M $(NH_4)_2SO_4$, 2 mM DTT, and injected on a Source[TM]15 Phe 4.6/100 PE column (Cytiva, Sweden). Elution was performed with a gradient ranging from 1 to 0 M $(NH_4)_2SO_4$ for 60 min at 0.5 ml.min$^{-1}$. *Mt*GS was eluted between 0.86 to 0.74 M $(NH_4)_2SO_4$. The final pool was diluted 1:100 in 50 mM Tricine/NaOH pH 8.0, 2 mM DTT, and loaded on a MonoQ[TM] 5/50 GL column (Cytiva, Sweden). *Mt*GS was eluted with a NaCl gradient ranging from 0 to 200 mM NaCl in 10 min followed by 200 to 500 mM NaCl in 60 min at 0.5 ml.min$^{-1}$ (eluted at 375-395 mM NaCl). The final pool was washed 1:1000 in 25 mM Tris/HCl pH 7.6, 10% glycerol, 150 mM NaCl, 2 mM DTT and concentrated to 200 μl with a 30-kDa cutoff concentrator (Sartorius, Germany). The protein concentration was estimated via Bradford assay, and the sample was flash-frozen and stored anaerobically at -80 °C.

*Ms*GS *purification*. The pellet (4 g) was suspended in 50 mM Tris/HCl pH 8.0, 2 mM DTT (lysis buffer). Cell lysis and preparation of extracts were performed similarly, except that the pellet after centrifugation was resuspended in the lysis buffer, sonicated and centrifuged a second time to extract additional proteins. Soluble fractions were pooled and diluted with the lysis buffer to obtain a final 15-fold dilution (final volume 60 ml). The filtered sample was loaded on a 2 × 5 mL HiTrap[TM] DEAE Sepharose FF (Cytiva,

Sweden) equilibrated with the same buffer. The protein was eluted with a 0 to 400 mM NaCl linear gradient for 150 min at a 2 ml.min$^{-1}$ flow rate. *Ms*GS eluted between 240 mM and 310 mM NaCl. The pooled sample was diluted with three volumes of lysis buffer and was loaded on a 5 ml HiTrap Q HP[TM] column (Cytiva, Sweden). The protein was eluted with a 150 to 500 mM NaCl linear gradient for 70 min at a 1 ml.min$^{-1}$ flow rate. *Ms*GS eluted between 0.38 and 0.41 M NaCl under these conditions. The pooled sample was diluted with 1 volume of 25 mM Tris/HCl pH 7.6, 2 M $(NH_4)_2SO_4$, 2 mM DTT before loading on a 5 ml HiTrap[TM] Phenyl Sepharose HP column (Cytiva, Sweden) equilibrated with the same buffer. *Ms*GS eluted with a 1.1 to 0 M $(NH_4)_2SO_4$ linear gradient for 70 min at a 1 ml.min$^{-1}$ flow rate. The protein eluted between 0.91 M and 0.81 M $(NH_4)_2SO_4$. Pooled fractions were concentrated on a 10-kDa cutoff centrifugal concentrator (Sartorius, Germany), and the buffer was exchanged for 25 mM Tris/HCl pH 7.6, 10% glycerol, 2 mM DTT.

**Size estimation of the GS**. The protocol of hrCN PAGE was adapted as described in Lemaire et al.[46] (originally described in Lemaire et al.[47]) and run at 40 mA for 1 h using 5-12% gradient gels. The size determination of *Mt*GS (540.3 kDa) and *Ms*GS (547.5 kDa) was obtained using a fit derived from the migration distances and sizes of the standard proteins.

Size exclusion chromatography was carried out under anaerobic conditions in a Coy tent filled with an $N_2/H_2$ atmosphere (97:3%), at 20 °C and under yellow light, using a Superose[TM] 6 Increase 10/300 GL column (Cytiva, Sweden). Chromatography was performed in 25 mM Tris/HCl buffer, pH 7.6, 10% glycerol, 2 mM DTT, at a flow rate of 0.4 ml.min$^{-1}$. 76 and 67.5 μg of *Mt*GS and *Ms*GS, respectively, were used. Size determination was performed using a fit derived from the elution volumes and sizes of the standard proteins.

**Activity assays**. GS activities were measured using the pyruvate kinase/ lactate dehydrogenase (PK/LDH) coupled enzymes from rabbit muscle ordered from Sigma-Aldrich (containing 600–1000 units.ml$^{-1}$ pyruvate kinase and 900–1400 units.ml$^{-1}$ lactate dehydrogenase). The activity was measured by following NADH oxidation resulting in the change of absorbance at 340 nm (Fig. S2).

Absorbance was measured aerobically in a 96-well plate with a SPEKTROstarNano (BMG Labtech, Germany) at 45 °C in 100 μl final reaction volume. GS was added immediately after opening the anaerobic storage flask to minimize the negative effects of oxygen and low temperature. Activities were performed with an enzyme coming from a single preparation and measured in technical triplicates.

All reagents were prepared in the reaction buffer (200 mM $KH_2PO_4$ pH 7, 10 mM KCl) apart from $MgCl_2 \times 6 H_2O$ which was dissolved in water. The standard reaction mix contained as a final concentration: 1 mM NADH (freshly prepared), 2 mM phosphoenolpyruvate (freshly prepared), 3 mM ATP (freshly prepared), 50 or 80 mM sodium glutamate (for *Mt*GS and *Ms*GS, respectively), 20 mM $NH_4Cl$, 25 mM $MgCl_2 \times 6H_2O$, 2 mM sodium 2OG (freshly prepared), 0.02 mg.ml$^{-1}$ GS and 5 μl PK/LDH from a 1:10 dilution of the stock. Kinetic parameters were determined by varying the glutamate concentrations from 0 to 100 mM for *Mt*GS and 0 to 200 mM for *Ms*GS, and $NH_4Cl$ from 0 to 5 mM for *Mt*GS and 0 to 20 mM for *Ms*GS. The determination of the $K_{0.5}$ and Hill coefficient for *Mt*GS activation was performed by varying the 2OG concentration from 0 to 5 mM. Glutamine feedback inhibition was measured at 20 mM glutamine. Unspecific interactions with malate and succinate for *Mt*GS were measured at 15 mM of the compounds by preparing

the reaction mix without 2OG, measuring for 20 min and then adding 2OG (2 mM final). $O_2$ sensitivity was measured in anaerobic cuvettes with 0.01 mg.ml$^{-1}$ $Mt$GS, starting the reaction with glutamate addition. All solutions were prepared anaerobically, and then half of the master mix, GS and glutamate were separately incubated for 1 h at ambient air. The anaerobic or aerobic compounds were combined, and activity was measured immediately. It has been previously observed that certain buffer conditions contribute to a faster activity loss, including compounds like β-mercaptoethanol or dithiothreitol[41]. Therefore, it should be considered when comparing specific activities from different studies, as it might also affect the $O_2$-effect on enzymes.

Rates were measured by exploiting the linear regression for each dataset through the points with the steepest, most linear slope. The control slope without the enzyme was subtracted from the slope obtained with the enzyme in the same timeframe. Activities are presented in μmol of NADH oxidized per minute per mg of added GS, using a molar extinction coefficient of $\varepsilon_{340nm} = 6220$ M$^{-1}$.cm$^{-1}$ for NADH. The apparent $K_m$ and $V_{max}$ were calculated with the $K_M$ $V_{max}$ Tool Kit (ic50tk/kmvmax.html).

**Crystallization.** All proteins were crystallized fresh without any freezing step, and all crystals were obtained through the sitting drop method on a 96-Well MRC 2-Drop Crystallization Plates in polystyrene (SWISSCI, UK) at 20 °C under anaerobic conditions ($N_2$/$H_2$, gas ratio of 97:3).

$Mt$GS apo without TbXo4 was crystallized at a concentration of 15 mg.ml$^{-1}$. The crystallization reservoir contained 90 μl of mother liquor (20% w/v polyethylene glycol 3,350 and 100 mM potassium sodium tartrate), the crystallization drop contained a mixture of 0.55 μl protein and 0.55 μl precipitant.. Hexagonal plates appeared after few weeks and were soaked in the mother liquor supplemented with 25% v/v ethylene glycol prior to freezing in liquid nitrogen.

$Mt$GS with 2OG/Mg$^{2+}$/ATP was crystallized at 3.7 mg.ml$^{-1}$ with a final concentration of 2 mM 2OG, 2 mM ATP and 2 mM MgCl. The crystallization reservoir contained 90 μl of mother liquor (20% w/v polyethylene glycol 3,350 and 200 mM sodium fluoride). The crystallization drop contained 0.6 μl GS with ligands and 0.6 μl precipitant. Short thick hexagonal rods appeared after few weeks and were soaked in the mother liquor supplemented with 20% v/v glycerol prior to freezing in liquid nitrogen.

$Mt$GS with 2OG/Mg$^{2+}$ was crystallized at 3.4 mg.ml$^{-1}$ with a final concentration of 2 mM 2OG and 2 mM MgCl. The crystallization reservoir contained 90 μl of mother liquor (25% w/v polyethylene glycol 1,500 and 100 mM SPG (succinic acid, sodium dihydrogen phosphate, and glycine) buffer pH 5.0), and the crystallization drop contained 0.6 μl GS with ligands and 0.6 μl precipitant. Hexagonal plates obtained after few weeks were soaked in the mother liquor supplemented with 15% v/v glycerol prior to freezing in liquid nitrogen.

All $Ms$GS crystals were obtained by mixing 0.55 μl protein at 9 mg.ml$^{-1}$ with 0.55 μl precipitant.

For $Ms$GS apo 1 the crystallization reservoir contained 90 μl of 200 mM ammonium formate and 20% (w/v) polyethylene glycol 3,350. Rectangular rods appeared within weeks, and crystals were soaked in the mother liquor supplemented with 25% v/v ethylene glycol prior to freezing in liquid nitrogen.

$Ms$GS-Mg$^{2+}$/ATP was obtained from the same condition. Here, crystals were soaked for 4 min in the crystallization solution containing ATP, MgCl$_2$ and sodium glutamate, each at a final concentration of 10 mM, and then back soaked in the crystallization solution supplemented with 25% glycerol prior to freezing in liquid nitrogen. It is worth noting that no additional electron density could be attributed despite soaking the $Ms$GS

crystal with 10 mM glutamate. Therefore, the structure was named $Ms$GS-Mg$^{2+}$/ATP bound complex. The absence of binding might come from a packing artifact, a low occupancy of magnesium in the active site, or the aberrant position of the ATP phosphate backbone.

For $Ms$GS apo 2 the crystallization reservoir contained 90 μl of 1.6 M sodium citrate tribasic dihydrate. Flat squares appeared within weeks, and crystals were soaked in the mother liquor supplemented with 30% v/v glycerol prior to freezing in liquid nitrogen.

**Data collection, processing and structure refinement.** All datasets were collected at 100 K at the Swiss Light Source (SLS), beamline PXIII (X06DA). Data processing was performed with $autoPROC$[48] combined with STARANISO[49] except for $Mt$GS apo co-crystallized with the TbXo4 (obtained from Engilberge et al.[31]), which was processed by $XDS$[50] and scaled with SCALA from the $CCP4$ package. The structure of $Mt$GS-apo-TbXo4 was solved by molecular replacement with MOLREP ($CCP4$)[51] by using the closest structural homolog from $Bacillus$ $subtilis$ ($Bs$GS resting state, PDB 4LNN). Other $Mt$GS structures were solved by molecular replacement with PHASER from the $PHENIX$ package[52] with the $Mt$GS-apo-TbXo4 model. $Mt$GS-2OG/Mg$^{2+}$/ATP was used as template to solve $Mt$GS-2OG/Mg$^{2+}$ and $Ms$GS-apo 1 structures with PHASER. The two other $Ms$GS structures were solved with PHASER by using $Ms$GS-apo 1 as template.

Refinement was performed with $PHENIX$[52] and $BUSTER$[53] in combination with fast automatic visual model building in $COOT$[54]. All models were systematically validated by using Molprobity[55]. $Mt$GS apo cocrystallized with TbXo4 was refined with considering all atoms except water anisotropic. Additionally, riding hydrogens were added during the refinement. $Mt$GS apo without TbXo4 was refined by applying translation libration screw model (TLS) and adding riding hydrogens. Hydrogens were omitted in these final deposited models. In addition, this dataset was refined by applying the following twin operator: 1/2 h + 1/2k,3/2h-1/2k,-l (twin fraction of 0.12). $Mt$GS-2OG/Mg$^{2+}$/ATP was refined in $BUSTER$ by applying non crystallography symmetry (NCS) and TLS, while $Mt$GS-2OG/Mg$^{2+}$ was refined in $PHENIX$ with TLS and without NCS and with riding hydrogens.

All $Ms$GS models were refined by applying TLS, without NCS and without generating hydrogens.

The PDB ID codes, data collection and refinement statistics for the deposited models are listed in Tables 2 and 3. The structures of $Mt$GS-apo without TbXo4 (PDB 8OON) and $Mt$GS-2OG/Mg$^{2+}$ (PDB 8OOQ) presented R-factors slightly higher than that averaged from structures of similar resolution, due to imperfections in the crystals. The refinement and electron density quality of the $Mt$GS-apo without TbXo4 structure was hampered by pseudomerohedral twinning. Numerous crystals were analyzed to remove or reduce the twinning, which was only obtained by adding the TbXo4 molecule[31]. In the present work, this structure is mainly used as a control to exclude an effect of TbXo4 on the protein. A total of 114 different crystals were analyzed in order to obtain the structure of $Mt$GS-2OG/Mg$^{2+}$, and the presented structure was obtained from the best dataset from this intensive screening and collection.

**Structural analyses.** All structure visualization was performed with PyMol (2.2.0 Schrödinger, LLC, New York, USA). rmsd putty graphs were generated by generating rmsd values with SUPERPOSE from the $CCP4$ package[51], substituting the B-values in the according PDB file with the new rmsd values and displaying it in the Putty preset in PyMol. Coloring was set to "spectrum b,

yellow_orange_magenta, minimum=0, maximum=5". Putty scaling was set to absolute linear scaling from the B-factor column.

When chain selection was necessary for structural alignment, the chain with the lowest average B-factor was used. The structures do not exhibit any major differences of B-factors between chains, and therefore similar conclusions would have been drawn with the selection of other chains for analysis.

The movie visualizing the conformational changes upon 2OG binding was generated with UCSF Chimera (Version 1.16, University of California, USA). A morph between *Mt*GS-apo-TbXo4 to *Mt*GS-2OG/Mg$^{2+}$ to *Mt*GS-2OG/Mg$^{2+}$/ATP is shown. The enzyme is displayed as cartoon with the ligands as spheres. 2OG carbons are colored yellow and ATP carbons light blue, while the other atoms are colored according to element (O in red, N in blue, P in orange). In the second half a zoom-in of the active site is shown with the Asp-50' loop in a darker blue and Arg321 is displayed as sticks. The two chains composing the active site are colored in different shades of cyan.

Sequence alignments were generated with Clustal Omega[56] and superposed to the secondary structure with the Esprit 3.0 server (https://esprit.ibcp.fr)[57].

The oligomerization state of all obtained structures was predicted via PDBePISA (Proteins, Interfaces, Structures and Assemblies, https://www.ebi.ac.uk/msd-srv/prot_int/cgi-bin/piserver). To facilitate processing, 8OOQ was processed as two separate dodecamers, and ligands were removed from 8OOO. The two best results for each structure are listed in Table S1.

**Phylogeny and residues conservation analysis**. The *Mt*GS sequence was used as query for a BLAST P[58] search in the RefSeq database, restricted to the *Archaea* domain. This database was used to limit the redundancy of sequences from similar species. The research limit was set to 500 sequences. The sequences of the GS from *B. subtilis*, *S. aureus*, *P. polymyxa* and *L. monocytogenes* were added, leading to a total of 504 sequences, all listed in Supplementary Data 2. The phylogenic tree was constructed by the MEGA X program[59]. The alignment was done by MUSCLE with default parameters and the tree was constructed using the Maximum Likelihood method and JTT matrix-based model[60]. The tree with the highest log likelihood (-191831.27) is shown. The node scores were calculated with 200 replications. There were a total of 656 positions in the final dataset. The branches were manually colored according to the taxonomy from the NCBI database[61]. The tree was visualized and imaged using iTol v6[62].

The sequences were manually separated in different monophyletic groups. The different sequences forming the groups were aligned by Clustal Omega[56] and the residue conservation images were constructed using Weblogo 3 (version 3.7.12)[63] based on the residues position in *Mt*GS and multiple sequences alignments.

**Reporting summary**. Further information on research design is available in the Nature Portfolio Reporting Summary linked to this article.

## Data availability

All protein models were deposited in the Protein data bank under the following PDB codes: 8OOL (*Mt*GS-apo-TbXo4), 8OON (*Mt*GS-apo without TbXo4), 8OOO (*Mt*GS-2OG/Mg$^{2+}$/ATP), 8OOQ (*Mt*GS-2OG/Mg$^{2+}$), 8OOW (*Ms*GS-apo 1), 8OOX (*Ms*GS-apo 2), 8OOZ (*Ms*GS-Mg$^{2+}$/ATP). Source data for the graphs in Fig. 1 and Fig. 5 can be found in Supplementary Data 1. Biochemical and crystallographic raw data will be made available on request.

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

## Acknowledgements

We would like to thank the Max Planck Institute for Marine Microbiology and the Max-Planck-Society for their continuous support. We are also grateful for the beam time allocation at SLS and the support of the PXIII beamline staff. We would also like to thank Christina Probian and Ramona Appel from the Microbial Metabolism laboratory for their continuous support, as well as Marion Jespersen and Nevena Maslać for providing *M. thermolithotrophicus* cells. We acknowledge Sylvain Engilberge, Éric Girard, Olivier Maury and François Riobé for their contribution to the *Ms*GS-TbXo4 structure. We are grateful to Stian Torset and Tomás Alarcón Schumacher for their help with data formatting for the rmsd graph generation. C.U.W. and J.M.K. were supported by the SIAM gravitation program (grant #024002002) granted by the Netherlands Organisation for Scientific Research and the Ministry of Education, Culture and Science. J.M.K was furthermore supported by the Deutsche Forschungs Gesellschafts (DFG) Grant KU 3768/1-1.

## Author contributions

*Methanothermococcus thermolithotrophicus* cultivation was mainly performed by M-C.M. *M. shengliensis* was cultivated by J.M.K. *Mt*GS and *Ms*GS were purified and crystallized by M-C.M and O.N.L., respectively. Biochemical characterization and activity assays were performed by M-C.M and O.N.L. X-ray data collection was performed by M-C.M., O.N.L and T.W. Data processing, model building, structure refinement, validation and deposition were performed by M-C.M and T.W. Structures were analyzed by M-C.M. and T.W. O.N.L. performed phylogenetic analyses. C.U.W. and T.W. acquired funding to realize the project. The paper was written by M-C.M, O.N.L. and T.W. with contributions and final approval of all co-authors.

## Funding

## Competing interests

The authors declare no competing interests.
