## [Peer Review File · Communications Biology]

Reviewers' comments:

Reviewer #1 (Remarks to the Author):

In their study, Müller et al. provide structural insights into metabolite-triggered regulatory mechanisms targeting glutamine synthetases (GS) of methanogenic archaea. The group of GS represented by the homolog from *Methanothermococcus thermolithotrophicus* (MtGS) needs to be activated by the binding of 2-oxoglutarat (2OG) to an allosteric binding site. Instead, the GS of *Methermicoccus shengliensis* (MsGS) is shown to be active irrespective of 2OG but sensitive to product inhibition by glutamine. From the comparison of the GS structures in different complexes with ATP and 2OG with the well characterized GS from *Bacillus subtilis* (BsGS) the authors suggest mechanisms through which both regulations occur. In the inactive resting state of MtGS the ATP binding site is blocked and R321 important for catalysis is retracted from the active site by the Asp50'-loop. 2OG binding induces a long-range conformational change making the active site accessible for ATP and transferring R321 into its catalytically active position. The feedback-inhibition by glutamine on the other hand, is known to be also mediated by the Asp50'-loop in the BsGS which harbours an arginine able to stabilize an interaction of a glutamate with the glutamine in the active site and thereby closing the active site. Because this arginine is conserved in the glutamine-sensitive MsGS but not in the glutamine-insensitive MtGS, the authors attribute the glutamine feedback inhibition to this residue. With these first structural analyses of archaeal homologs another important piece could be added to the knowledge on the highly diverse GS regulation. By elucidating the biochemical mechanism and its attribution to specific amino acids, a good prediction of archaeal GS regulation based on sequence analyses is enabled. The manuscript is clearly written and the important claims are conclusively proven by the presented data. I only noted some minor points that should be considered by the authors.

I. 107/122 and figure 1A: It is notable, that in the SDS-PAGE as well as in the hrCN-PAGE the MsGS bands appear to present a larger protein compared to the MtGS, although the MtGS is stated to be the larger protein. Please double-check the protein sizes and the captions of the lanes. Additionally, it is not understandable why the MsGS band in the SDS-PAGE was estimated to ~48kDa in the text considering the protein size of 49 kDa. It would be more comprehensible to state the height of the band as coherent with the size of the protein.

I. 112-113: Later in the discussion the same concentrations of 2OG are all given in mM. May the authors unify the units of 2OG to mM also here.

I. 116: It is argued that an activation of MtGS by 2OG is still possible also after adding malate or succinate. Figure S.3 proves this claim. However, the activity is clearly lower than with just malate or succinate. Please consider mentioning the decrease of activity.

I. 126-128 and figure 1E: The authors state that the apparent KM values of MtGS and MsGS are similar. However, they differ by two-fold for ammonia and four-fold for glutamate. Please rephrase to "in the same order of magnitude" (or sth. similar). Additionally, it is not clearly described, if the activity with increasing substrate concentrations shows Michaelis-Menten-like behaviour. E.g. for some GS homologs a substrate inhibition by glutamate was shown (Deuel and Stadtman, 1970). Referring to the substrate saturation curves used for the calculation of kinetic parameters in the supplements would enhance comprehensibility.

I. 181-185: The authors discuss the steric effect of the substitution of V94' in the allosteric 2OG binding pocket of MtGS by E89' in MsGS and E36' and Y20' in BsGS. In my point of view, it is noteworthy that in both cases a potentially negatively charged amino acid is inserted in the binding pocket. An additional ionic repulsion could be discussed here.

I. 205: An additional reference to figure 5 at this point could help to understand the changes in the

active site induced by the long-range effect of 2OG binding and the similarities of this state to the BsGS complex state.

Figure 5F: The first bar for the specific activity of MsGS is not labelled. Therefore, it is not clear if 2OG was added in these measurements.

Reviewer #2 (Remarks to the Author):

The manuscript by Muller et al is a very interesting and generally well written document describing analyses of the archaeal, *Methanothermococcus thermolithotrophicus* (Mt) and *Methermicoccus shengliensis* (Ms), glutamine synthetases and effects of small molecule regulators. Specifically, the authors perform biochemistry and obtain crystal structures of the GS in apo, Mg, 2-oxoglutarate (2-OG) (for Mt) bound and ATP bound forms. They show biochemically and structurally that Mt GS binds 2-OG. Their enzyme assays show, in fact, this interaction is required for Mt GS activity while this is not the case for Ms GS. Instead Ms GS is regulated similar to low G+C Gram-positive GS by binding the product, glutamine. Based on these data, in particular their structural analyses, they propose that these- not mutually exclusive -modes of regulation may be at least partially predictable in archaeal GS. I am supportive of this study but have a few, minor, issues that should be addressed prior to publication.

1. A central point of this study is the binding of 2-OG and its impact on Mt GS function. They should include in the main manuscript, Sigma-A-weighted mFo-DFc difference omit electron density for the 2-OG.
2. Instead of the bar graphs (figure 1D) the authors should use graphs that include the individual data points. And include if replicates were technical or biological replicates.
3. It is difficult to see the interactions with 2-OG in Figure 3 A (and B-C showing why there would not be interactions). The authors should remake the figure to make this important interaction clearer.
4. Page 7, line 200-201, the authors indicate the opening observed during comparing the Mt GS apo and 2OG bound is similar to the apo and "complex" of *B. subtilis*. They need to add a reference here as it is unclear what the comparison is referring to ie. It is not stated what *B. subtilis* "complex" (Transition state?) they are referring to for the *B. subtilis* structure.
5. What does ".. excluding the sequestration of an artefactual conformation provoked by the compound" (TbXo4) mean? Can the authors rewrite this sentence.

Reviewer #3 (Remarks to the Author):

1. The primary focus of Muller et al.'s study is on the structural investigation of glutamine synthetase (GS) from two archaeal species, MtGS and MsGS. The authors present structures of these archaeal GS enzymes in distinct catalytic states, including apo, Mg²⁺-bound, and complex forms. Notably, these are the first reported GS structures from Archaea. While the structures of MtGS and MsGS are generally similar to known bacterial GS, there are some differences in the structural details. The authors conduct a comprehensive comparison of their structures with BsGS, drawing several noteworthy conclusions. For instance, the ATP-bound form of MsGS is found to be catalytically inactive, similar to the reported behavior of BsGS. Overall, I found this manuscript to be engaging and

commend the authors for their thorough investigation of GS structures and enzymatic activities across different species.

2. In the manuscript, the resolution of the structures is often described in a non-standard format, using "number" + "-" + "Å" (e.g., 1.65-Å on page 5, line 136). It might be more conventional to use "1.65 Å" instead of "1.65-Å." Although there are numerous instances of this description, the authors might consider adopting the standard format for consistency, unless it is a formal requirement.

3. I noticed that the structural qualities (R_{work} , R_{free}) of 800N and 800Q structures appear to be comparatively worse than those determined at the same resolution. It would be valuable if the authors could discuss any efforts made to improve the structures' quality or provide a few sentences describing the quality of structures used for analysis and comparison.

4. The manuscript lacks clear evidence regarding the oligomeric states of the two studied GS enzymes in solution. Since the crystal structures sometimes show different oligomeric forms (e.g., dimer, hexamer, or dodecamer), I recommend performing additional experiments like AUC or SEC-MALS to confirm the assembly, at least for the apo-form. Given recent findings of GS forming tetradecamers in some species, such evidence in solution is critical to ensure the crystal structures are not influenced by crystal packing. Additionally, further analysis demonstrating the ring-ring interactions in the crystal units and providing biophysical evidence for the 12-mer assembly of MtGS and MsGS would be beneficial.

5. The residue positions of MtGS in alignments shown in Figures 3D, 4D, 5E, and S7 are easily discernible. However, the same cannot be said for MsGS and other GS enzymes in the aligned sequences (positions). I recommend enhancing the clarity of these alignments to facilitate better residue position analysis.

6. I suggest that the enzymatic kinetic curves/data used in 1D and 1E be provided, similar to the supporting data for 1C. Since the two GS versions exhibit distinct Hill coefficients, it would be valuable for the authors to elaborate on this finding. For example, does it imply that the dodecameric GS sometimes exhibits an auto-inhibited form (e.g., 30% of the time), while the remainder remains active?

7. The authors have compared MtGS, an Archaeal GS, with the Gram-positive bacterial GS, BsGS. It would be helpful to include comparisons or analyses with GS from Gram-negative bacteria, such as *E. coli* GS and *Salmonella typhimurium* GS, to provide a more comprehensive understanding of GS variations across different bacterial species.

8. On page 5, line 160, the authors are encouraged to include the alignment RMSD (Root Mean Square Deviation) number, as it was not found in the main text or figure legend.

9. In several instances on lines 363, 366, and 367, the authors mention the ionic state of NaH_2PO_4 , while using NaPO_4^{-2} in some contexts. Similarly, the authors use $(\text{NH}_4)_2\text{SO}_4$ without specifying the ionic state for the solute. To enhance clarity, it would be helpful to consistently state the ionic state of the solutes throughout the manuscript.

10. Figures 1F, S5A, S6A, S8A, S9A, and S12 could benefit from indicating the "aligned" monomer using arrows to assist readers in understanding the figures better. Some of the aligned monomers are clear, but others may require additional guidance.

11. Fig S4A and Fig S10A/C should include a gradient bar and range of B-factor to illustrate the differences. If the monomeric subunits do not share the same B-factors, the authors should specify the variations.

12. In Fig S10, panels A and C represent the B-factors, not A and B as described in the legend. Clarifying this discrepancy would be beneficial for readers.

Response to the Reviewers' comments

We thank the reviewers for their time, insightful comments, and suggestions. The point-by-point responses can be found below, with the according modifications underlined for clarity.

Reviewers' comments:

Reviewer #1 (Remarks to the Author):

In their study, Müller *et al.* provide structural insights into metabolite-triggered regulatory mechanisms targeting glutamine synthetases (GS) of methanogenic archaea. The group of GS represented by the homolog from *Methanothermococcus thermolithotrophicus* (MtGS) needs to be activated by the binding of 2-oxoglutarat (2OG) to an allosteric binding site. Instead, the GS of *Methermicoccus shengliensis* (MsGS) is shown to be active irrespective of 2OG but sensitive to product inhibition by glutamine. From the comparison of the GS structures in different complexes with ATP and 2OG with the well characterized GS from *Bacillus subtilis* (BsGS) the authors suggest mechanisms through which both regulations occur. In the inactive resting state of MtGS the ATP binding site is blocked and R321 important for catalysis is retracted from the active site by the Asp50'-loop. 2OG binding induces a long-range conformational change making the active site accessible for ATP and transferring R321 into its catalytically active position. The feedback-inhibition by glutamine on the other hand, is known to be also mediated by the Asp50'-loop in the BsGS which harbours an arginine able to stabilize an interaction of a glutamate with the glutamine in the active site and thereby closing the active site. Because this arginine is conserved in the glutamine-sensitive MsGS but not in the glutamine-insensitive MtGS, the authors attribute the glutamine feedback inhibition to this residue.

With these first structural analyses of archaeal homologs another important piece could be added to the knowledge on the highly diverse GS regulation. By elucidating the biochemical mechanism and its attribution to specific amino acids, a good prediction of archaeal GS regulation based on sequence analyses is enabled. The manuscript is clearly written and the important claims are conclusively proven by the presented data. I only noted some minor points that should be considered by the authors.

1. 107/122 and figure 1A: It is notable, that in the SDS-PAGE as well as in the hrCN-PAGE the MsGS bands appear to present a larger protein compared to the MtGS, although the MtGS is stated to be the larger protein. Please double-check the protein sizes and the captions of the lanes. Additionally, it is not understandable why the MsGS band in the SDS-PAGE was estimated to ~48kDa in the text considering the protein size of 49 kDa. It would be more comprehensible to state the height of the band as coherent with the size of the protein.

(Our answer) Indeed, the molecular weight of both proteins determined experimentally by SDS and hrCN PAGE does not reflect the expected calculated ones. Yet, we verified the caption and samples several times. Therefore, we attributed the difference to an artifact from the electrophoretic migration.

In contrast, the elution profiles of both GS on size exclusion chromatography are coherent, with MsGS exhibiting an estimated Stokes radius smaller than MtGS.

We clarified the text by adding a sentence and the result from the size exclusion chromatography:

*Lines 123-131

“*MsGS* was anaerobically purified from *M. shengliensis* leading to a major band at ~50 kDa on SDS PAGE (Fig. 1A). A dodecameric assembly was suggested by hrCN PAGE (estimated complex size at 547.5 kDa, Fig. 1B). Both the SDS and hrCN PAGE indicate a molecular weight for *MsGS* (WP_042685700.1, predicted molecular weight: 49.53 kDa) superior to that of *MtGS* (WP_018154487.1, predicted molecular weight: 50.24 kDa), a difference that we attributed to an artifact from the electrophoretic migration. When subjected to size exclusion chromatography, both proteins exhibited an elution volume in the range of the proposed dodecameric assembly in which *MsGS* is smaller than *MtGS* (Fig. S4). Native electrophoresis and size exclusion chromatography indicate a single oligomeric state for the purified GS.”

l. 112-113: Later in the discussion the same concentrations of 2OG are all given in mM. May the authors unify the units of 2OG to mM also here.

The units were unified in the whole manuscript, including the mentioned section.

*Lines 112-113

“No activity was detected below 0.06 mM 2OG, 50% activity was reached at 0.17 mM and saturation occurred above ~ 0.6 mM.”

l. 116: It is argued that an activation of *MtGS* by 2OG is still possible also after adding malate or succinate. Figure S.3 proves this claim. However, the activity is clearly lower than with just malate or succinate. Please consider mentioning the decrease of activity.

The decreased activity of the 2OG-activated enzyme in the presence of malate/succinate is now mentioned in the main text.

*Lines 115-118

“Specificity for 2OG was tested by substitution with malate or succinate, which are structurally similar to 2OG. At 15 mM, both surrogates could not activate *MtGS*. Malate or succinate addition to the 2OG-activated enzyme slightly impaired the activity, an effect that might be due to a competition for the binding site (Fig. S3).”

l. 126-128 and figure 1E: The authors state that the apparent K_M values of *MtGS* and *MsGS* are similar. However, they differ by two-fold for ammonia and four-fold for glutamate. Please rephrase to “in the same order of magnitude” (or sth. similar). Additionally, it is not clearly described, if the activity with increasing substrate concentrations shows Michaelis-Menten-like behaviour. E.g. for some GS homologs a substrate inhibition by glutamate was shown (Deuel and Stadtman, 1970). Referring to the substrate saturation curves used for the calculation of kinetic parameters in the supplements would enhance comprehensibility.

“similar” was substituted with “the same order of magnitude” as suggested.

*Line 134-135

“Both enzymes exhibit the same order of magnitude for the apparent K_m for glutamate and ammonium,…”

Kinetic curves for both substrates and enzymes were added as supplementary figure S5.

l. 181-185: The authors discuss the steric effect of the substitution of V94' in the allosteric 2OG

binding pocket of MtGS by E89' in MsGS and E36' and Y20' in BsGS. In my point of view, it is noteworthy that in both cases a potentially negatively charged amino acid is inserted in the binding pocket. An additional ionic repulsion could be discussed here.

Thanks for this comment. The ionic repulsion of 2OG through negatively charged residues in MsGS and BsGS is now discussed in the manuscript.

*Lines 193-195

“In the 2OG-insensitive MsGS structure, the side chain of Glu89' substituting the MtGS Val94' would collide with the 2OG position, and its negative charge would additionally lead to the ionic repulsion of 2OG (Fig. 3B).”

*Lines 196-198

“Similarly, the structures of the bacterial homologs show a tyrosine and glutamate at the equivalent positions 18' and 34' in MtGS that would collide with the modeled 2OG and would contribute to ionic repulsion (Fig. 3C).”

l. 205: An additional reference to figure 5 at this point could help to understand the changes in the active site induced by the long-range effect of 2OG binding and the similarities of this state to the BsGS complex state.

The additional reference to Fig. 5 was added. (Now line 218)

Figure 5F: The first bar for the specific activity of MsGS is not labelled. Therefore, it is not clear if 2OG was added in these measurements.

The according label was added for clarification.

Reviewer #2 (Remarks to the Author):

The manuscript by Muller et al is a very interesting and generally well written document describing analyses of the archaeal, *Methanothermococcus thermolithotrophicus* (Mt) and *Methermicoccus shengliensis* (Ms), glutamine synthetases and effects of small molecule regulators. Specifically, the authors perform biochemistry and obtain crystal structures of the GS in apo, Mg, 2-oxoglutarate (2-OG) (for Mt) bound and ATP bound forms. They show biochemically and structurally that Mt GS binds 2-OG. Their enzyme assays show, in fact, this interaction is required for Mt GS activity while this is not the case for Ms GS. Instead Ms GS is regulated similar to low G+C Gram-positive GS by binding the product, glutamine. Based on these data, in particular their structural analyses, they propose that these- not mutually exclusive -modes of regulation may be at least partially predictable in archaeal GS. I am supportive of this study but have a few, minor, issues that should be addressed prior to publication.

1. A central point of this study is the binding of 2-OG and its impact on Mt GS function. They should include in the main manuscript, Sigma-A-weighted mFo-DFc difference omit electron density for the 2-OG.

The sigma-A-weighted mF_o-DFc difference omit electron densities of 2OG for the two structures, is now presented in Fig. S15. We placed the new picture in supplementary rather than as a main figure because the configuration of Figure 3 in the revised version is already overcrowded due to the stereo image of panel A (see point number 3 from this referee). Additionally we added the mF_o-DFc difference omit electron densities of ATP in Figure S18.

2. Instead of the bar graphs (figure 1D) the authors should use graphs that include the individual data points. And include if replicates were technical or biological replicates.

The individual data points were added to the bar graphs in Fig. 1D, Fig. 5F, and Fig. S3. Additionally, it was specified that all activities were measured in technical triplicates in the material and methods section and the figure legends.

*Lines 432-433

“Activities were performed with an enzyme coming from a single preparation and measured in technical triplicates.”

*Lines 769-770

“(E) Kinetic parameters of both GS for NH_4Cl and glutamate. All activities were measured in technical triplicates. The curves used for kinetics parameter determination are provided in Fig. S5.”

*Line 819

“(F) Specific activity in the absence and presence of glutamine in both archaeal GS ($n=3$).”

3. It is difficult to see the interactions with 2-OG in Figure 3 A (and B-C showing why there would not be interactions). The authors should remake the figure to make this important interaction clearer.

The figure was modified as suggested. The figure 3A now includes a stereo view of the binding site, and background elements were removed in Figures 3A-C.

4. Page 7, line 200-201, the authors indicate the opening observed during comparing the Mt GS apo and 2OG bound is similar to the apo and “complex” of *B. subtilis*. They need to add a reference here as it is unclear what the comparison is referring to ie. It is not stated what *B. subtilis* “complex” (Transition state?) they are referring to for the *B. subtilis* structure.

The compared state of *BsGS* was specified.

*Lines 213-214

“The observed switch and the 2OG-bound active state are similar to the one observed between apo and transition state complex of *B. subtilis* (Fig. S16).”

5. What does “.. excluding the sequestration of an artefactual conformation provoked by the compound” (TbXo4) mean? Can the authors rewrite this sentence.

The sentence was rephrased as follows.

*Lines 156-159

“The protein structure (*MtGS*-apo-without TbXo4) was refined to 2.43 Å and is almost identical to the TbXo4 -containing form (RMSD of 0.206 Å, 410 atoms aligned Fig. S8, residues 67-69 could not be modeled in the TbXo4 -lacking structure), confirming that the observed conformation is not the artefactual result of the TbXo4 binding.”

Reviewer #3 (Remarks to the Author):

1. The primary focus of Muller et al.'s study is on the structural investigation of glutamine synthetase (GS) from two archaeal species, *MtGS* and *MsGS*. The authors present structures of these archaeal GS enzymes in distinct catalytic states, including apo, Mg^{2+} -bound, and complex forms. Notably, these are the first reported GS structures from Archaea. While the structures of *MtGS* and *MsGS* are generally similar to known bacterial GS, there are some differences in the structural details. The authors conduct a comprehensive comparison of their structures with *BsGS*, drawing several noteworthy conclusions. For instance, the ATP-bound form of *MsGS* is found to be catalytically inactive, similar to the reported behavior of *BsGS*. Overall, I found this manuscript to be engaging and commend the authors for their thorough investigation of GS structures and enzymatic activities across different species.

2. In the manuscript, the resolution of the structures is often described in a non-standard format, using "number" + "-" + "Å" (e.g., 1.65-Å on page 5, line 136). It might be more conventional to use "1.65 Å" instead of "1.65-Å." Although there are numerous instances of this description, the authors might consider adopting the standard format for consistency, unless it is a formal requirement.

The spelling of resolutions was standardized.

3. I noticed that the structural qualities (R_{work} , R_{free}) of 8OON and 8OOQ structures appear to be comparatively worse than those determined at the same resolution. It would be valuable if the authors could discuss any efforts made to improve the structures' quality or provide a few sentences describing the quality of structures used for analysis and comparison.

The relatively bad structural quality of the structures 8OON and 8OOQ is due to the intrinsic quality of the crystals reflected by the diffraction experiment.

The structure 8OON exhibits pseudomerohedral twinning with a twin fraction of 0.12, which impairs the statistics. This twinning issue is the reason for using the crystallophore TbXo4 that solved the problem. This structure is a control to exclude an effect of TbXo4 on the overall protein conformation and was not used to describe the domain arrangements and active site as explained in the text.

The structure 8OOQ was solved using the diffraction data obtained after screening 114 different crystals, the best being kept for structure determination. The asymmetric unit contains two *MtGS*, an assembly of circa 1.2 MDa that was challenging to refine at this resolution and taking into account the non-negligible translational non-crystallography symmetry. Yet statistically worse than the structure of a similar resolution, the presented structure is of enough quality for the performed structural analysis, which was mainly a control to assess that the *MtGS* conformational switch is impacted by 2OG and not ATP.

A discussion of the quality of these structures was added in the material and methods section.

*Lines 515-522

“The structures of *MtGS*-apo without TbXo_4 (PDB 8OON) and *MtGS*-2OG/ Mg^{2+} (PDB 8OOQ) presented R-factors slightly higher than that averaged from structures of similar resolution due to imperfections in the crystals. The refinement and electron density quality of the *MtGS*-apo without TbXo_4 structure was hampered by pseudomerothedral twinning. Numerous crystals were analyzed to remove or reduce the twinning, which was only obtained by adding the TbXo_4 molecule³¹. In the present work, this structure is mainly used as a control to exclude an effect of TbXo_4 on the protein. A total of 114 different crystals were analyzed in order to obtain the structure of *MtGS*-2OG/ Mg^{2+} , and the presented structure was obtained from the best dataset from this intensive screening and collection.”

4. The manuscript lacks clear evidence regarding the oligomeric states of the two studied GS enzymes in solution. Since the crystal structures sometimes show different oligomeric forms (e.g., dimer, hexamer, or dodecamer), I recommend performing additional experiments like AUC or SEC-MALS to confirm the assembly, at least for the apo-form. Given recent findings of GS forming tetradecamers in some species, such evidence in solution is critical to ensure the crystal structures are not influenced by crystal packing. Additionally, further analysis demonstrating the ring-ring interactions in the crystal units and providing biophysical evidence for the 12-mer assembly of *MtGS* and *MsGS* would be beneficial.

The different crystalline forms presented in this work exhibit a different content of monomers in the asymmetric units. Dodecamers have been systematically observed in these different crystalline forms that have been obtained from different precipitants and even at different pH. The PDBePISA webserver tool predicts a stable dodecameric assembly from all presented structures (results are gathered in supplementary Table S3) and confirms that the dodecameric state is more stable compared to the hexamer, illustrating the importance of the ring-ring contact to maintain the oligomerization. Therefore, we are confident about the dodecameric state of both proteins in the crystals.

Supplementary figures were added to illustrate the conservation of the dodecameric state in the crystalline packing and clarify the asymmetric unit content (Table S3, Fig. S6, and S10).

An additional sentence was added in the material and methods section.

*Lines 541-544

“The oligomerization state of all obtained structures was predicted via PDBePISA (Proteins, Interfaces, Structures and Assemblies, https://www.ebi.ac.uk/msd-srv/prot_int/cgi-bin/piserver). To facilitate processing, 8OOQ was processed as two separate dodecamers, and ligands were removed from 8OOO. The two best results for each structure are listed in Table S3.”

Regarding the state of the enzyme in solution, high-resolution clear native polyacrylamide gel electrophoresis indicates a single population of GS, with a molecular weight coherent with the dodecameric assembly seen in crystalline packing that has now been quantified. The multiplicity of crystalline forms is also in favor of the dodecamerisation, which would not be a crystallization artifact. Hence, at least a population of both proteins in solution is dodecameric. To verify that both protein samples are homogenous and fit with a dodecameric state in solution, we performed analytical size exclusion chromatography. We found a unique population of high molecular weight (estimated to be 531.3 kDa for *MtGS* and 505.7 kDa for *MsGS*), which is in accordance with our previous results.

For the above reasons, we are fairly confident that the purified enzymes should be in a dodecameric state. However, we cannot conclude that the enzymes would keep this oligomerization in a physiological environment where protein partners or cellular regulators might influence it, which is out of the scope of the present work.

The quantification from the high-resolution clear native polyacrylamide gel electrophoresis is stated in the main text and materials and methods. The size exclusion chromatography experiment is reported in Figure S4 and added in the main text.

*Lines 106-108

“Denaturing and high-resolution clear native polyacrylamide gel electrophoresis (hrCN PAGE) are coherent with a dodecameric organization of the ~50 kDa peptide (complex size estimated at 540.3 kDa, Fig. 1A and B).”

*Lines 123-131

“*MsGS* was anaerobically purified from *M. shengliensis* leading to a major band at ~50 kDa on SDS PAGE (Fig. 1A). A dodecameric assembly was suggested by hrCN PAGE (estimated complex size at 547.5 kDa, Fig. 1B). Both the SDS and hrCN PAGE indicate a molecular weight for *MsGS* (WP_042685700.1, predicted molecular weight: 49.53 kDa) superior to that of *MtGS* (WP_018154487.1, predicted molecular weight: 50.24 kDa), a difference that we attributed to an artifact from the electrophoretic migration. When subjected to size exclusion chromatography, both proteins exhibited an elution volume in the range of the proposed dodecameric assembly in which *MsGS* is smaller than *MtGS* (Fig. S4). Native electrophoresis and size exclusion chromatography indicate a single oligomeric state for the purified GS.”

*Lines 415-424

“The protocol of hrCN PAGE was adapted as described in Lemaire et al. 47 (originally described in Lemaire et al. 48) and run at 40 mA for 1 h using 5-12% gradient gels. The size determination of *MtGS* (540.3 kDa) and *MsGS* (547.5 kDa) was obtained using a fit derived from the migration distances and sizes of the standard proteins.”

“Size exclusion chromatography was carried out under anaerobic conditions in a Coy tent filled with an N₂/H₂ atmosphere (97:3 %), at 20 °C and under yellow light, using a SuperoseTM 6 Increase 10/300 GL column (Cytiva, Sweden). Chromatography was performed in 25 mM Tris/HCl buffer, pH 7.6, 10% glycerol, 2mM DTT, at a flow rate of 0.4 ml.min⁻¹. 76 and 67.5 µg of *MtGS* and *MsGS*, respectively, were used. Size determination was performed using a fit derived from the elution volumes and sizes of the standard proteins.”

An additional discussion about the effect of regulators on the oligomeric state was added.

*Lines 342-347

“In contrast, the sP26 protein described in *M. mazei* is not encoded in the genomes of *M. thermolithotrophicus* and *M. shengliensis*, and the same applies to the repressor GlnR forming higher oligomer species in the bacterial system. However, non-homologous functional equivalents may exist. Similarly, if our experiments do not suggest any modification of the oligomeric state, such a regulatory mechanism could occur in the cell in the presence of other regulatory partners, which might become separated during the purification process.”

5. The residue positions of *MtGS* in alignments shown in Figures 3D, 4D, 5E, and S7 are easily discernible. However, the same cannot be said for *MsGS* and other GS enzymes in the aligned

sequences (positions). I recommend enhancing the clarity of these alignments to facilitate better residue position analysis.

While we agree with the point made, adding residue numbers for all aligned sequences in the main figure for each organism will interfere with figure readability and will lead to an overcrowding of the main figures. As an alternative, we added residue numbering for all sequences to the full alignment found in Supplementary Figure S11, allowing readers to find residue positions for each protein sequence.

6. I suggest that the enzymatic kinetic curves/data used in 1D and 1E be provided, similar to the supporting data for 1C. Since the two GS versions exhibit distinct Hill coefficients, it would be valuable for the authors to elaborate on this finding. For example, does it imply that the dodecameric GS sometimes exhibits an auto-inhibited form (e.g., 30% of the time), while the remainder remains active?

Kinetic curves for both substrates and both enzymes were added in the supplementary Fig. S5.

Regarding the question on the Hill coefficients, we believe that the hill coefficient for *MtGS* regarding Glutamate and ammonium is marginal compared to the one observed for *MsGS* for glutamate. Both enzymes, assumed to be in a dodecameric state in the purified sample, would differ in their cooperativity regarding glutamate, yet the biochemical determinants of this difference in cooperativity are not known.

The text was modified to reflect the data and emphasize the difference between both enzymes:

*Lines 134-139

“Both enzymes exhibit the same order of magnitude for the apparent K_m for glutamate and ammonium, and binding of both substrates exhibits a marginal positive cooperativity in *MtGS*, while in *MsGS* glutamate binding showed a positive cooperativity and almost no cooperativity for ammonium. Such observed differences in cooperativity between both enzymes might hide a more sophisticated discrepancy in their mode of regulation and sensing the intracellular metabolite balance.”

7. The authors have compared *MtGS*, an Archaeal GS, with the Gram-positive bacterial GS, *BsGS*. It would be helpful to include comparisons or analyses with GS from Gram-negative bacteria, such as *E. coli* GS and *Salmonella typhimurium* GS, to provide a more comprehensive understanding of GS variations across different bacterial species.

Since all structurally studied Bacterial GS (apart from the members of the order *Bacillales*, which we mention in the manuscript) do not belong to Group I- α , which is the focus of the present work, we did not include them. The comparison with other types of GS that exhibit different regulation strategies would probably be less meaningful for the scope of the study and increase the risk of impairing comprehension.

8. On page 5, line 160, the authors are encouraged to include the alignment RMSD (Root Mean Square Deviation) number, as it was not found in the main text or figure legend.

RMSD values were added according to the reviewer's suggestion.

*Lines 170-173

“A structural comparison of the monomeric unit reveals that *MsGS* has a closer fit with *BsGS* compared to *MtGS* (RMSD *MsGS-BsGS*: 0.749 Å, 341 atoms aligned, *MtGS-BsGS*: 0.847 Å, 338 atoms aligned), which exhibits larger deviations such as an extended helix α_2 resulting from a seven-residue insertion (Fig 2G, Fig. S7).”

9. In several instances on lines 363, 366, and 367, the authors mention the ionic state of NaH_2PO_4 , while using NaPO_4^{2-} in some contexts. Similarly, the authors use $(\text{NH}_4)_2\text{SO}_4$ without specifying the ionic state for the solute. To enhance clarity, it would be helpful to consistently state the ionic state of the solutes throughout the manuscript.

The text was modified according to the reviewer's suggestion.

10. Figures 1F, S5A, S6A, S8A, S9A, and S12 could benefit from indicating the "aligned" monomer using arrows to assist readers in understanding the figures better. Some of the aligned monomers are clear, but others may require additional guidance.

An indication for the superposed subunit was added to all figures with superpositions containing multiple subunits, according to the reviewer's suggestion.

11. Fig S4A and Fig S10A/C should include a gradient bar and range of B-factor to illustrate the differences. If the monomeric subunits do not share the same B-factors, the authors should specify the variations.

Gradient bars for the B-factor were added to the supplementary figures.

When necessary, the chain with the lowest average B-factor was used for comparison between structures, yet no significant or notable difference in B-factor within chains could be seen. The selection of the chain with the lowest average B-factor for analysis was added in the text and figure legend.

*Lines 529-531

“When chain selection was necessary for structural alignment, the chain with the lowest average B-factor was used. The structures do not exhibit any significant differences of B-factors between chains, and therefore similar conclusions would have been drawn with the selection of other chains for analysis.”

12. In Fig S10, panels A and C represent the B-factors, not A and B as described in the legend. Clarifying this discrepancy would be beneficial for readers.

The legend was corrected.

Additional comments:

During the reviewing process, we noticed incorrect values in Fig 1E, and a minor adjustment was performed. Our interpretation is unchanged. Additionally, we slightly adjusted the according section to more accurately describe the corrected data.

*Lines 134-139

“Both enzymes exhibit the same order of magnitude for the apparent K_m for glutamate and ammonium, and binding of both substrates exhibits a marginal positive cooperativity in *MtGS*, while in *MsGS* glutamate binding showed a positive cooperativity and almost no cooperativity for ammonium. Such observed differences in cooperativity between both enzymes might hide a more sophisticated discrepancy in their mode of regulation and sensing the intracellular metabolite balance.”

REVIEWERS' COMMENTS:

Reviewer #3 (Remarks to the Author):

The authors have adeptly addressed all of my comments and queries, providing clear and comprehensive responses. The revised manuscript, enriched with additional information, effectively elucidates their research. The figures are presented in a reader-friendly manner, enhancing the overall clarity. I am thoroughly satisfied with the current version and wholeheartedly recommend its publication in *Communications Biology*.